# The complex of TRIP-Br1 and XIAP ubiquitinates and degrades multiple adenylyl cyclase isoforms

Wenbao Hu[1†], Xiaojie Yu[1†], Zhengzhao Liu[1†], Ying Sun[1], Xibing Chen[1], Xin Yang[1], Xiaofen Li[2], Wai Kwan Lam[1], Yuanyuan Duan[2‡], Xu Cao[1], Hermann Steller[3], Kai Liu[1,4], Pingbo Huang[1,2,4*]

[1]Division of Life Science, Hong Kong University of Science and Technology, Hong Kong, China; [2]Division of Biomedical Engineering, Hong Kong University of Science and Technology, Hong Kong, China; [3]Strang Laboratory of Apoptosis and Cancer Biology, Howard Hughes Medical Institute, The Rockefeller University, New York, United States; [4]State Key Laboratory of Molecular Neuroscience, Hong Kong University of Science and Technology, Hong Kong, China

**Abstract** Adenylyl cyclases (ACs) generate cAMP, a second messenger of utmost importance that regulates a vast array of biological processes in all kingdoms of life. However, almost nothing is known about how AC activity is regulated through protein degradation mediated by ubiquitination or other mechanisms. Here, we show that transcriptional regulator interacting with the PHD-bromodomain 1 (TRIP-Br1, Sertad1), a newly identified protein with poorly characterized functions, acts as an adaptor that bridges the interaction of multiple AC isoforms with X-linked inhibitor of apoptosis protein (XIAP), a RING-domain E3 ubiquitin ligase. XIAP ubiquitinates a highly conserved Lys residue in AC isoforms and thereby accelerates the endocytosis and degradation of multiple AC isoforms in human cell lines and mice. XIAP/TRIP-Br1-mediated degradation of ACs forms part of a negative-feedback loop that controls the homeostasis of cAMP signaling in mice. Our findings reveal a previously unrecognized mechanism for degrading multiple AC isoforms and modulating the homeostasis of cAMP signaling.

**\*For correspondence:**
bohuangp@ust.hk

[†]These authors contributed equally to this work

**Present address:** [‡]Biomedical Translational Research Institute, Jinan University, Guangzhou Shi, China

**Competing interests:** The authors declare that no competing interests exist.

## Introduction

cAMP, the first second messenger discovered by Earl Sutherland in 1957, plays a crucial role in regulating a vast array of biological processes in all kingdoms of life by targeting PKA, Epac, cyclic nucleotide-gated ion channels, and Popdc proteins. cAMP is produced from ATP by adenylyl cyclases (ACs). Based on their structural and functional similarities, 9 transmembrane ACs (tmACs) in mammals are classified into four groups (*Patel et al., 2001*). All tmACs share the common feature of containing two catalytic domains and an N-terminal domain. All tmACs except AC9 are activated by both the heterotrimeric G protein subunit Gsα and forskolin. Furthermore, the distinct AC isoforms are regulated differentially by myriad molecules, including Giα, Gβγ, calmodulin, PKA, PKC, protein associated with Myc (PAM), AKAP79/150, Snapin, and RGS2 (*Patel et al., 2001*; *Wang et al., 2009*). Surprisingly, almost all previous studies on ACs have focused on the regulation of their activities per se, and despite the pivotal role of ACs in cAMP signaling, next to nothing is known about the degradation of cell-surface ACs mediated by ubiquitination and/or other mechanisms that acutely and finely tune the signaling potential of cell-surface membrane proteins in response to environmental cues (*Piper and Luzio, 2007*; *MacGurn et al., 2012*).

TRIP-Br1 belongs to a recently discovered protein family that includes TRIP-Br1 (SEI-1, SERTAD1), TRIP-Br2 (SEI-2, SERTAD2), TRIP-Br3 (SEI-3, HEPP, CDCA4), and RBT1 (SERTAD3) in mammals and TARA in *Drosophila* (*Lai et al., 2007*; *Zang et al., 2009*). The functions of TRIP-Br1 and other members of its family have not been fully characterized. Although earlier studies suggested that TRIP-Br1 in the nucleus acts as a transcriptional co-regulator of E2F-responsive genes, or forms an active quaternary complex with cyclin D/CDK4/INK4a to promote normal cell cycle progression (*Sugimoto et al., 1999*; *Hsu et al., 2001*), recent studies showed that endogenous TRIP-Br1 is predominantly localized in the cytoplasm and undergoes regulated nucleo-cytoplasmic transport (*Zang et al., 2009*; *Hong et al., 2009*; *Jung et al., 2013*; *Hong et al., 2011*; *Lee et al., 2009*). Cytoplasmic TRIP-Br1 bound to 2 E3 ligases, XIAP and NEDD4-1, and prevented their ubiquitination and degradation (*Hong et al., 2009*; *Jung et al., 2013*), whereas it induced the ubiquitination and degradation of ASK1 (apoptosis signal-regulating kinase 1) and PKC-δ (*Hong et al., 2011*; *Lee et al., 2009*). However, physical interaction between TRIP-Br1 and ASK1/PKC-δ has not yet been observed (*Hong et al., 2011*; *Lee et al., 2009*). XIAP, NEDD4-1, and ASK1 are predominantly cytoplasmic, but PKC-δ is localized in both the cytoplasm and the nucleus.

## Results

### TRIP-Br1 binds to AC1

Our preliminary data indicated that exogenous TRIP-Br1 may interact with AC1. Further study suggested that endogenous TRIP-Br1 and AC1 coprecipitated from HeLa cell extracts (*Figure 1a*), and a GST-TRIP-Br1 fusion protein—but not GST alone—pulled down HA-tagged AC1 from HEK293T cells (*Figure 1b*). In addition, AC1 colocalized with TRIP-Br1 in HeLa cells, even in a super-resolution microscopy analysis (*Figure 1—figure supplement 1*). Next, we mapped the interaction sites in both TRIP-Br1 and AC1. TRIP-Br1 amino acids 50–82, which contain the SERTA domain, appeared to be required for binding AC1 (*Figure 1b*), whereas AC1 amino acids 236–612 (AC1-M), which include catalytic domain 1 (C1; aa 293–612), bound to TRIP-Br1 in pairwise pull-down assays (*Figure 1c–d*), indicating a direct binding of AC1-M with TRIP-Br1. Further mapping showed that catalytic domain 1b (C1b; aa 494–612) of AC1 is sufficient for binding TRIP-Br1 (*Figure 1c and e*).

### TRIP-Br1 promotes AC1 degradation

To assess the functional impact of TRIP-Br1 binding to AC1, we first examined AC1 protein expression in the presence and absence of TRIP-Br1. Stable expression of TRIP-Br1-V5 markedly reduced the total and cell-surface protein levels of endogenous AC1 in HEK293T cells (*Figure 2a*), but exerted little effect on other membrane proteins such as endogenous transferrin receptor (*Figure 2a*) and exogenous TRPV4 and TRPP2 (*Figure 2—figure supplement 1*). Agreeing with this observation, stable expression of TRIP-Br1-V5 also substantially reduced the protein level and forskolin-stimulated cAMP production of exogenous AC1 in HEK293T cells (Figure 6b–d). Conversely, knocking down TRIP-Br1 with two different shRNAs elevated AC1 protein expression in HeLa cells (*Figure 2b*). Furthermore, knocking out TRIP-Br1 in mice increased AC1 protein expression by 2–4 folds in heart tissue, adipocytes, and the brain (*Figure 2c*) without affecting AC1 mRNA levels (*Figure 2—figure supplement 1c*), which indicates that TRIP-Br1 regulates AC1 expression through a non-transcriptional function. Although TRIP-Br1 in the brain was too scarce to detect using our anti-TRIP-Br1 antibody, it greatly impacted AC1 protein expression (*Figure 2c*).

To explore how TRIP-Br1 reduces AC1 protein expression, we added the protein-synthesis blocker cycloheximide (CHX) to HeLa cells. Knocking down TRIP-Br1 considerably slowed down the degradation of endogenous AC1 (*Figure 2d*). The degradation of endogenous AC1, seemed to be mainly through a proteasomal pathway, because proteasomal but not lysosomal inhibitors significantly elevated AC1 expression (*Figure 2e*).

### TRIP-Br1 bridges the interaction of AC1 with XIAP E3 ligase

How does TRIP-Br1 promote AC1 degradation? Because TRIP-Br1 reportedly binds to XIAP (*Hong et al., 2009*), a RING E3 ligase, we tested whether TRIP-Br1 recruits XIAP to AC1 and thereby promotes AC1 ubiquitination and degradation. First, we verified the interaction between TRIP-Br1 and XIAP by using GST pull-down assays. GST-TRIP-Br1 fusion protein but not GST alone captured

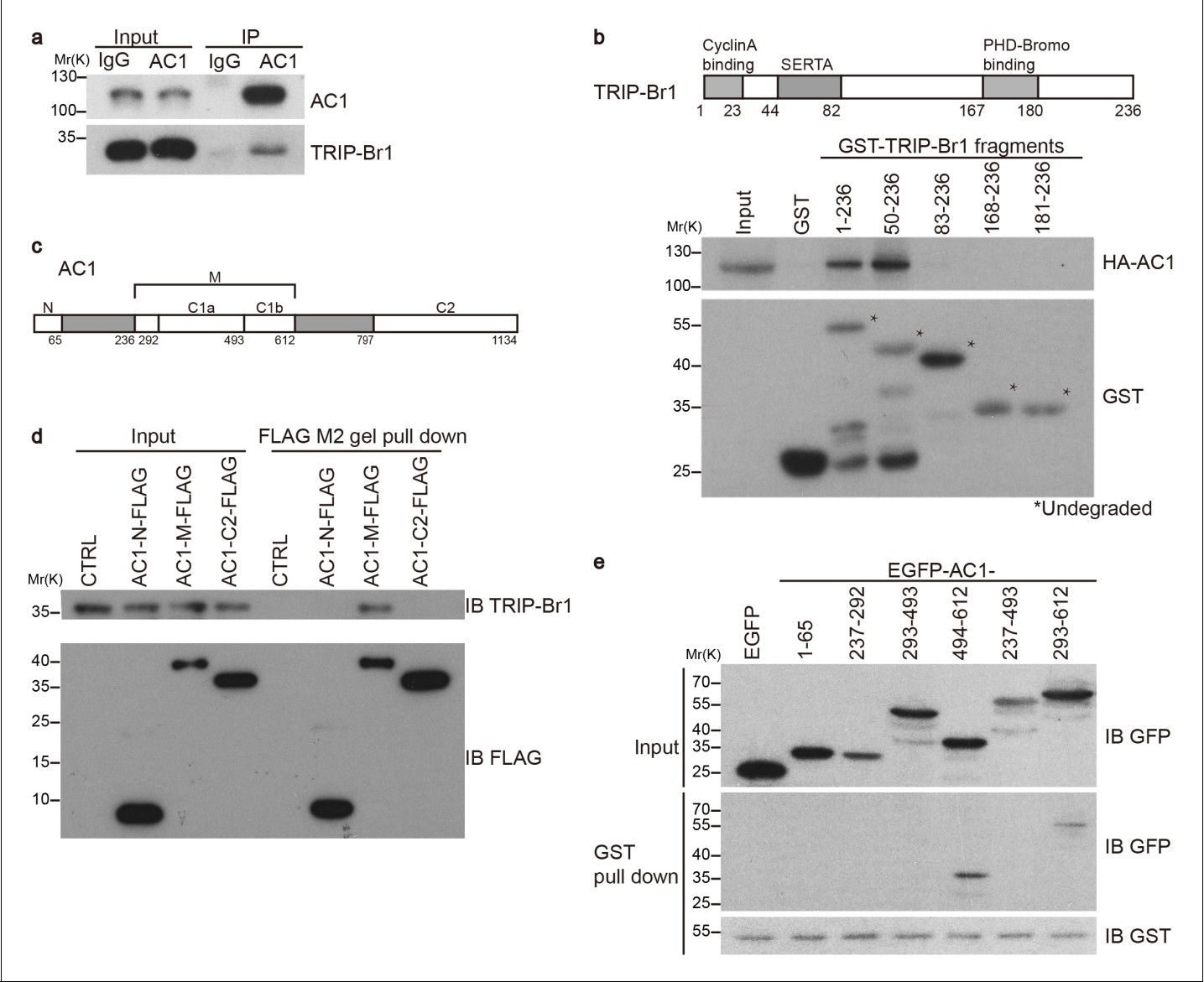

**Figure 1.** TRIP-Br1 interacts with AC1. (a) Endogenous AC1 in HeLa cells was immunoprecipitated (IP) with anti-AC1 antibody or control IgG and immunoblotted with anti-AC1 (top) and anti-TRIP-Br1 antibodies (bottom). (b) GST-TRIP-Br1 truncation mutants (bottom) were used to pull down HA-AC1 (middle) expressed in HEK293T cells. The domains included in the GST-TRIP-Br1 truncation mutants are illustrated in the schematic of TRIP-Br1 (top). *undegraded GST-TRIP-Br1 fragments. (c) Schematic of AC1. N, N-terminus; C1 and C2, catalytic domains 1 and 2, respectively; filled areas, transmembrane domains; M, cytosolic region between the 2 transmembrane domains. (d) Three purified FLAG-His-tagged AC1 fragments, AC1-N, AC1-M, and AC1-C2 (bottom), were used to pull down purified TRIP-Br1 (His-tagged at both N and C termini, top). CTRL, control: bovine serum albumin used instead of FLAG-His-tagged AC1 fragments. (e) GST-TRIP-Br1 was used to pull down GFP-tagged AC1 N-terminus and truncation fragments of C1 domain expressed in HEK293T cells. All experiments shown here are representative of 3–5 independent experiments.

The following figure supplement is available for figure 1:

**Figure supplement 1.** Colocalization of AC1 with TRIP-Br1.

XIAP purified from *Escherichia coli*, which demonstrated a potential direct interaction between XIAP and TRIP-Br1 (*Figure 3a*). Furthermore, we mapped the XIAP binding site in TRIP-Br1 to the first 49 amino acids of TRIP-Br1 (*Figure 3a*). By contrast, purified FLAG-His-tagged AC1-N, AC1-M, and AC1-C2 (*Figure 1c*) did not coprecipitate with purified XIAP, which indicated a lack of direct interaction between XIAP and AC1 (*Figure 3b*). However, FLAG-His-tagged AC1-M coprecipitated with

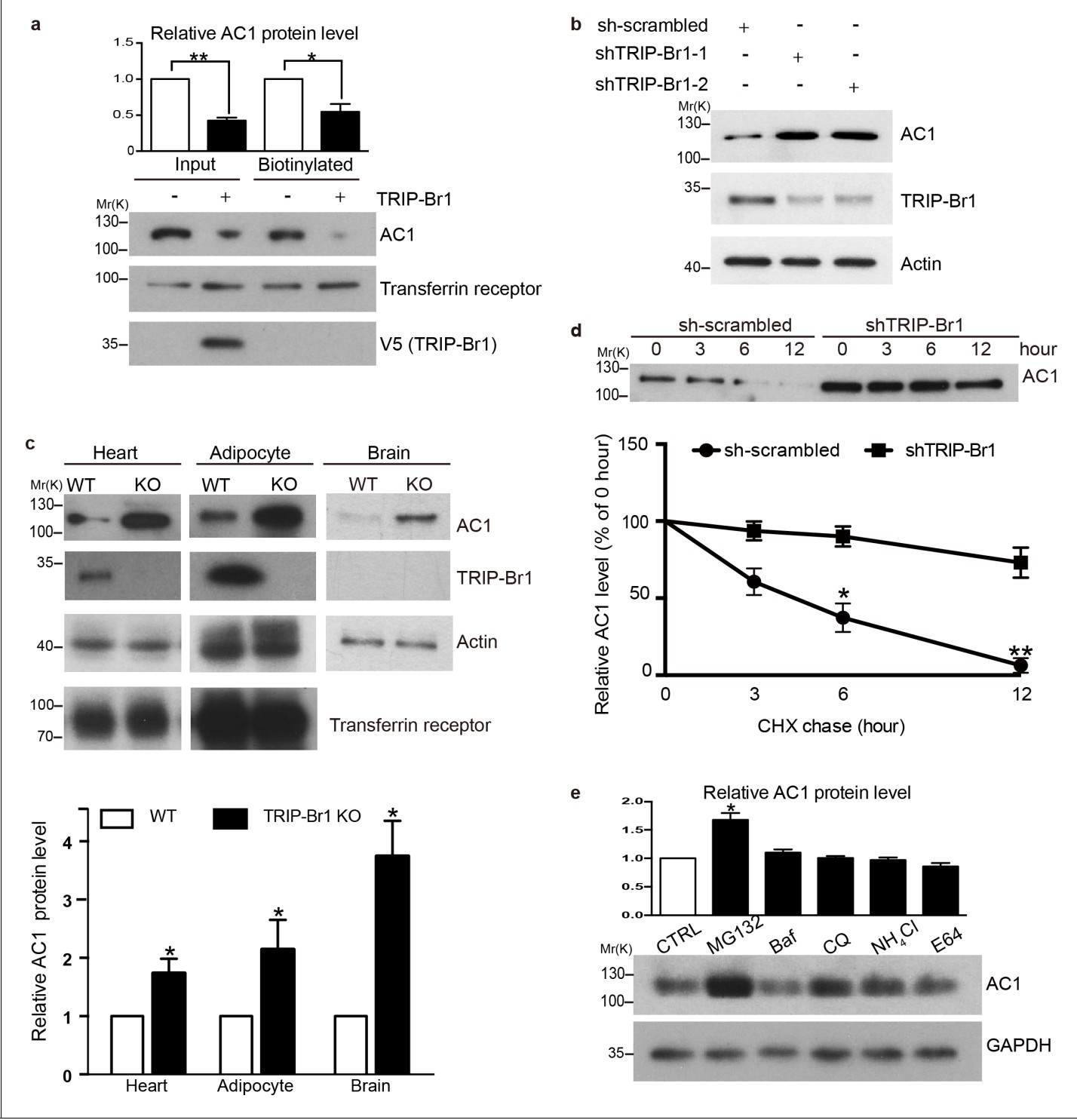

**Figure 2.** TRIP-Br1 promotes AC1 degradation. (a) Stable expression of TRIP-Br1-V5 markedly reduced total and cell-surface protein levels of endogenous AC1 in HEK293T cells. Cell-surface AC1 was biotinylated and isolated. Transferrin receptor: loading control. Quantification of the western blots is shown at the top: **, different from the control cell (CTRL), p=0.0053; *p=0.048; n = 3 independent experiments. (b) Knocking down TRIP-Br1 with two different shRNAs (shTRIP-Br1-1 and shTRIP-Br1-2) increased AC1 protein expression in HEK293T cells. Actin: loading control. (c) Knocking out TRIP-Br1 in mice elevated AC1 expression in heart tissue (n = 8, p=0.01), and adipocytes (n = 4, p=0.049), and the brain (n = 3, p=0.011). (d) Changes in the expression of endogenous AC1 with or without shTRIP-Br1-1 treatment were examined for 12 hr after CHX treatment in HEK293T cells (upper). The results are quantified in the lower panel; *, different from control (CTRL), p=0.013 (6 h); **p=0.004 (12 h); n = 3. (e) Effect of proteasomal and lysosomal

*Figure 2 continued on next page*

*Figure 2 continued*

inhibitors on endogenous AC1 in HEK293T cells. Baf, bafilomycin A1; CQ, chloroquine diphosphate. Quantification of the western blots is shown at the top: *, different from the control (CTRL), p=0.029; n = 3 independent experiments.

The following figure supplement is available for figure 2:

**Figure supplement 1.** Effect of TRIP-Br1 on the expression of other membrane proteins and AC1 mRNA levels.

purified XIAP in the presence of TRIP-Br1 (*Figure 3b*), which strongly suggested that TRIP-Br1 bridges the interaction between XIAP and AC1. This is further supported by the co-immunoprecipitation experiments showing that endogenous AC1, TRIP-Br1, and XIAP form a macromolecular complex in HeLa cells (*Figure 3c*).

We next tested whether XIAP binding results in AC1 ubiquitination and degradation and whether this process requires the presence of TRIP-Br1. Overexpression of wild-type XIAP, but not its H467A

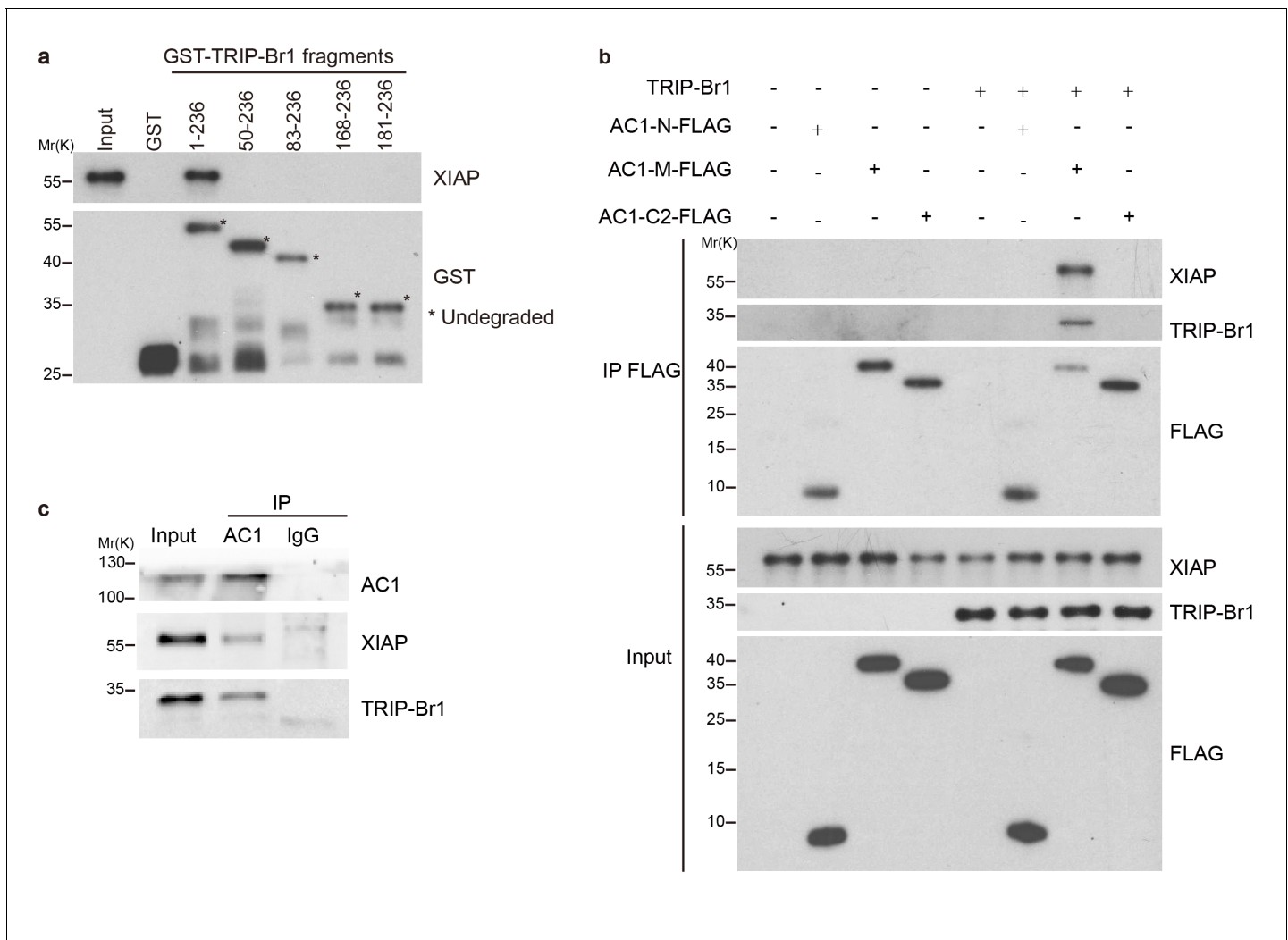

**Figure 3.** TRIP-Br1 bridges the interaction of AC1 with XIAP E3 ligase. (a) Full-length TRIP-Br1 fused with GST—but not GST-TRIP-Br1 truncation mutants or GST alone—captured His-XIAP purified from *E. coli*. (b) Three purified FLAG-His-tagged AC1 fragments, AC1-N, AC1-M, and AC1-C2, were used to pull down purified XIAP in the presence or absence of purified TRIP-Br1 (His-tagged at both N and C termini). (c) Endogenous AC1 in HeLa cells was IPed with anti-AC1 antibody or control IgG and immunoblotted with anti-AC1, anti-XIAP, and anti-TRIP-Br1 antibodies.

and F495A mutants that lack E3 ligase activity (*Yang et al., 2000*; *Damgaard et al., 2012*), markedly reduced the expression of endogenous AC1 and increased its ubiquitination in HeLa cells (*Figure 4a*), implying that XIAP E3 ligase directly or indirectly targets AC1. Moreover, overexpression of TRIP-Br1-FLAG lowered the expression of endogenous AC1 in HeLa cells as in HEK293T cells (*Figure 2*), and also elevated the ubiquitination of endogenous AC1 (*Figure 4a*); this suggested that TRIP-Br1 promotes the ubiquitination and degradation of AC1. Conversely, knockdown of XIAP or TRIP-Br1 with 2 or 3 different shRNAs produced the opposite effect of XIAP or TRIP-Br1 overexpression on endogenous AC1, respectively (*Figure 4b–c*, *Figure 4—figure supplement 1*, and *Figure 2b*). More importantly, the effect of XIAP on AC1 was almost abolished following extensive knockdown of endogenous TRIP-Br1 (*Figure 4b*), and vice versa (*Figure 4c*), and this suggested that XIAP and TRIP-Br1 collaboratively ubiquitinate and degrade AC1. Furthermore, in in vitro ubiquitination assays performed using only purified proteins, AC1 ubiquitination by XIAP required the presence of TRIP-Br1 (*Figure 5c*). Significantly, deleting the RING domain of XIAP in mice, in addition to increasing XIAP expression (*Schile et al., 2008*), markedly increased AC1 protein expression in heart tissue, adipocytes, and the brain, which clearly demonstrated that the E3 ligase activity of XIAP modulates AC1 expression in vivo (*Figure 4d*). In addition, deleting the RING domain of XIAP abolished the effect of TRIP-Br1 knockout in the mouse brain (*Figure 4e*), further fortifying the notion that XIAP and TRIP-Br1 act in concert (*Figure 4b–c*). On the basis of all these results, we conclude that TRIP-Br1 acts as an adaptor for the ubiquitination and degradation of AC1 by XIAP E3 ligase.

Among the IAP (inhibitor of apoptosis) family proteins, both cIAP1 and cIAP2 are closely related to XIAP and possess E3 ligase activity and a BIR2 domain. However, cIAP1 and cIAP2 did not or only weakly bound to TRIP-Br1(*Figure 4—figure supplement 2a*), reflecting that their BIR2 domains lack the 3 amino acid residues critical for TRIB-Br1 binding (*Hong et al., 2009*). Moreover, neither cIAP1 nor cIAP2 had any significant effect on AC1 protein expression (*Figure 4—figure supplement 2b*), suggesting that XIAP, among the three closely related IAP family proteins, specifically targets ACs.

## The conserved residue K1047 is the XIAP-mediated ubiquitin-conjugation site in AC1

Because TRRP-Br1 binds to the C1b domain of AC1, which shares high sequence similarity with the corresponding domains in other tmAC isoforms, we reasoned that TRIP-Br1 could potentially bind to all tmAC isoforms. Interestingly, our preliminary data suggested that TRIP-BR1 binds to multiple tmAC isoforms and TRIP-Br1 and XIAP appeared to promote the degradation of multiple AC isoforms (also see *Figures 6–7*). Thus, we reasoned that the sequences flanking the XIAP-mediated ubiquitination site(s) must be highly conserved in the distinct isoforms of AC. Based on sequence alignment (*Figure 6e*), 7 Lys residues flanked by highly conserved sequences in the cytoplasmic regions in multiple AC isoforms were identified and individually mutated to Arg in AC1. Notably, the K1047R mutation completely suppressed XIAP-induced AC1 degradation in HEK293T cells, whereas the other 6 mutations exerted negligible effects (*Figure 5a*). Moreover, the K1047R mutation almost eliminated XIAP-induced ubiquitination of AC1 (*Figure 5b*).

To verify that XIAP catalyzes the ubiquitination of AC1 directly instead of through another E3 ligase, we assembled an in vitro ubiquitination system that contained the E1 ubiquitin-activating enzyme (Uba1), the E2 ubiquitin-conjugating enzyme (UbcH5b), ATP, ubiquitin, and the protein substrate (AC1 or its K1047R and K1049R mutants). The addition of the E3 ubiquitin ligase XIAP led to the ubiquitination of AC1 and its K1049R mutant but not its K1047R mutant, but this ubiquitination occurred only in the presence of TRIP-Br1 (*Figure 5c*); these results clearly demonstrated that XIAP directly ubiquitinates AC1 and conjugates ubiquitins at the K1047 residue in a TRIP-Br1-dependent manner.

The pattern of XIAP-induced ubiquitination of AC1 appeared to be polyubiquitination (*Figure 5*), and an analysis of the linkage type of the polyubiquitination suggested that it is predominantly K27-linked (*Figure 5—figure supplement 1*). K27-linked polyubiquitination has been implicated in proteasomal or lysosomal degradation of proteins (*Birsa et al., 2014*; *Xu et al., 2009*; *Xia et al., 2013*; *Potapova et al., 2009*; *Birsa et al., 2014*).

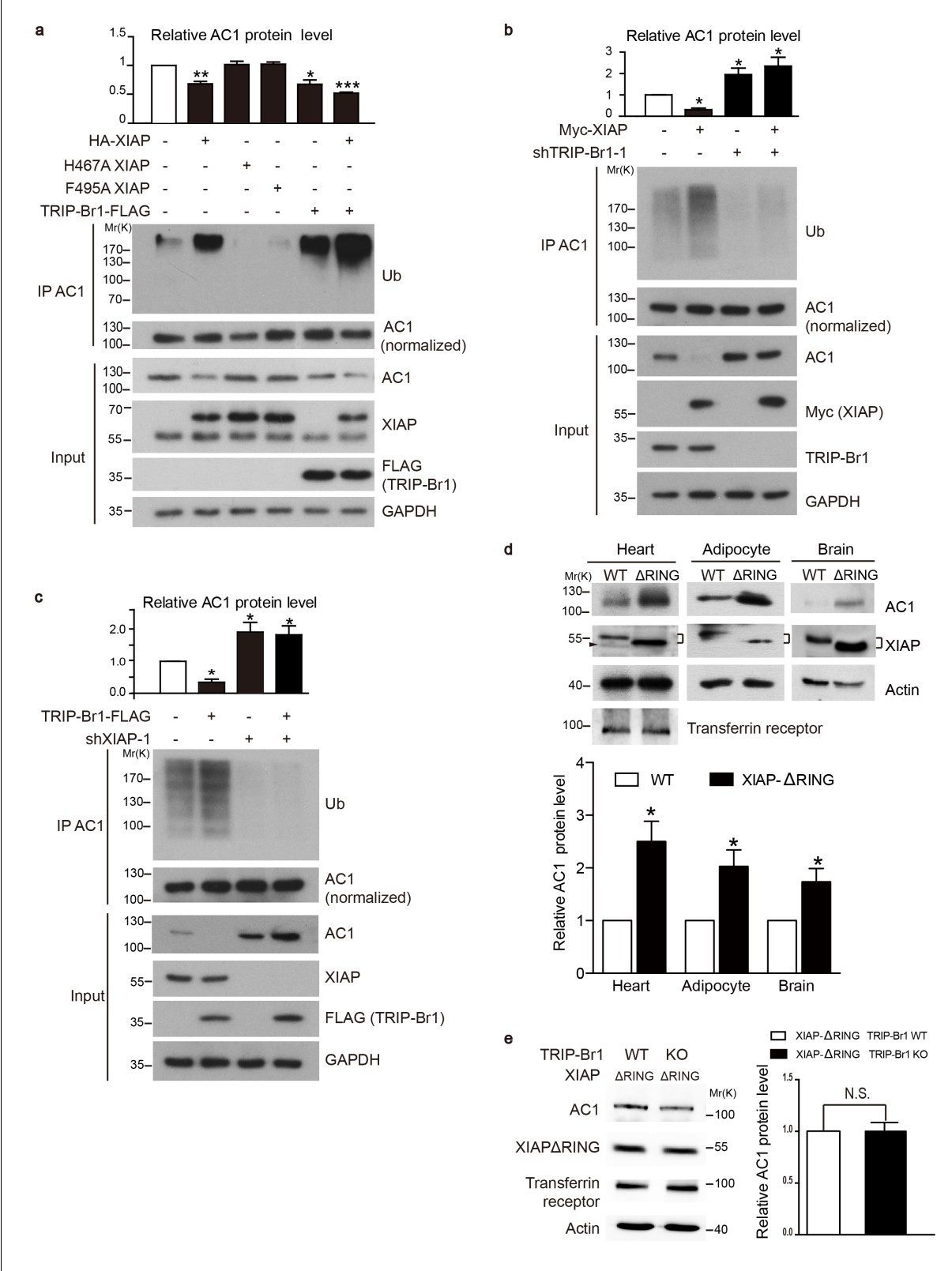

**Figure 4.** XIAP binding results in AC1 ubiquitination and degradation in a TRIP-Br1 dependent manner. (a–c) Ubiquitinated and total endogenous AC1 were examined following overexpression of HA-XIAP or its 2 mutants (H467A and F495A) and TRIP-Br1-FLAG in control HeLa cells (a) and in HeLa cells in which we knocked down TRIP-Br1 (b) or XIAP (c). shTRIP-Br1-1 and shXIAP-1, small hairpin RNA; Ub, ubiquitin; GAPDH, loading control. Loading of immunoprecipitated AC1 (or other AC isoforms) was normalized (adjusted to equal amounts) to clearly reveal the differences in ubiquitination of AC1

*Figure 4 continued on next page*

*Figure 4 continued*

(or other AC isoforms) in this and other figures. Total endogenous AC1 levels in panels **a–c** are quantified in their respective top bar graphs: *, different from control, p≤0.041; **p=0.0019; ***p<0.0001; n = 3 for all three panels. (d) Deleting the RING domain in XIAP in mice elevated AC1expression in heart tissue, adipocytes, and the brain. *, different from control, p≤0.045, n = 4 for all 3 tissues. Connected dashes, XIAP; arrowhead, nonspecific band. (e) Deleting the RING domain in XIAP in mice eliminated TRIP-Br1's effect on AC1 in the brain. N.S., not different from TRIP-Br1, p=0.943, n = 4. In panels **d** and **e**, actin, loading control; transferrin receptor, membrane-protein negative control.

The following figure supplements are available for figure 4:

**Figure supplement 1.** Effect of two other XIAP shRNAs on AC1 protein expression.

**Figure supplement 2.** cIAP1 and cIAP2 do not affect AC1 expression.

## XIAP/TRIP-Br1-mediated ubiquitination accelerates AC1 endocytosis

TRIP-Br1 reduced the surface expression of AC1 and accelerated its degradation (*Figure 2a and d*); therefore, we tested whether TRIP-Br1-induced ubiquitination increases the endocytosis of cell-surface AC1 or hinders the recycling of AC1 from intracellular pools. We examined the effect of TRIP-Br1 and XIAP on the endocytosis and recycling rates of exogenous wild-type and K1047R AC1 in HeLa cells. Expression of both TRIP-Br1 and XIAP robustly increased the endocytic rate of cell-surface AC1 but exerted no effect on the recycling of intracellular AC1 (*Figure 5—figure supplement 2*); more importantly, the K1047R mutation of AC1, which eliminates AC1 ubiquitination induced by TRIP-Br1/XIAP (*Figure 5b–c*), almost abolished the endocytosis of cell-surface AC1 in the absence and presence of TRIP-Br1 and XIAP (*Figure 5—figure supplement 2*). These results clearly demonstrate that TRIP-Br1/XIAP-induced ubiquitination of cell-surface AC1 accelerates its endocytosis and thereby increases its degradation.

## XIAP and TRIP-Br1 control the expression of multiple AC isoforms

Because TRIP-Br1 bound to the C1b domain in AC1, which is highly conserved among distinct AC isoforms, we hypothesized that TRIP-Br1 also binds to other AC isoforms. Our results showed that GST-TRIP-Br1 pulled down all five other AC isoforms tested: AC2, AC4, AC5, AC6, and AC8 (*Figure 6*). More importantly, the protein levels of these exogenous AC isoforms were also substantially diminished following TRIP-Br1 expression in HEK293T cells, as shown by the results of both western blotting and cAMP assays (*Figure 6b–d*).

Among these 5 AC isoforms, AC2 and AC5/6 were chosen for further examination of XIAP's involvement in their degradation, because AC1 (*Figures 1–6*), AC2, and AC5/6 represent 3 distinct classes of AC isoforms (*Patel et al., 2001*). Because of their high homology, AC5 and AC6 cannot be differentiated using currently available antibodies. As in the case of AC1 (*Figure 4a*), expression of XIAP but not its H467A and F495A mutants and expression of TRIP-Br1-FLAG reduced the protein expression of endogenous AC2 and AC5/6 and increased the ubiquitination of enxogenous AC2 and AC5 (*Figure 7a–c*, and *Figure 7—figure supplement 1a*); this supports the conclusion that AC2 and AC5/6, like AC1, are substrates of the TRIP-Br1/XIAP E3 ligase complex. Conversely, knockdown of XIAP or TRIP-Br1 produced the opposite effect of XIAP or TRIP-Br1 overexpression on AC2 and AC5/6 (*Figure 7—figure supplement 1b–c*). Moreover, the effect of XIAP on AC2 and AC5/6 was almost abolished following extensive knockdown of endogenous TRIP-Br1, and vice versa (*Figure 7—figure supplement 1b–c*). These results indicated that XIAP and TRIP-Br1 also collaboratively ubiquitinate and degrade AC2 and AC5/6.

We also determined that when K1065 in AC2 and K1247 in AC5, which are both aligned with K1047 in AC1, were mutated to Arg, XIPA/TRIP-Br1-induced ubiquitination and degradation of AC2 and AC5 were eliminated (*Figure 7b–c*); this further bolstered the notion that XIAP/TRIP-Br1-mediated ubiquitination is a common mechanism for degrading multiple AC isoforms.

In agreement with the aforementioned results, knockout of TRIP-Br1 or deletion of the RING domain of XIAP (XIAP ΔRING) significantly increased expression of AC2 and AC5/6 in heart tissue and the brain in mice (*Figure 7d–e* and *Figure 7—figure supplement 2*). Collectively, our data

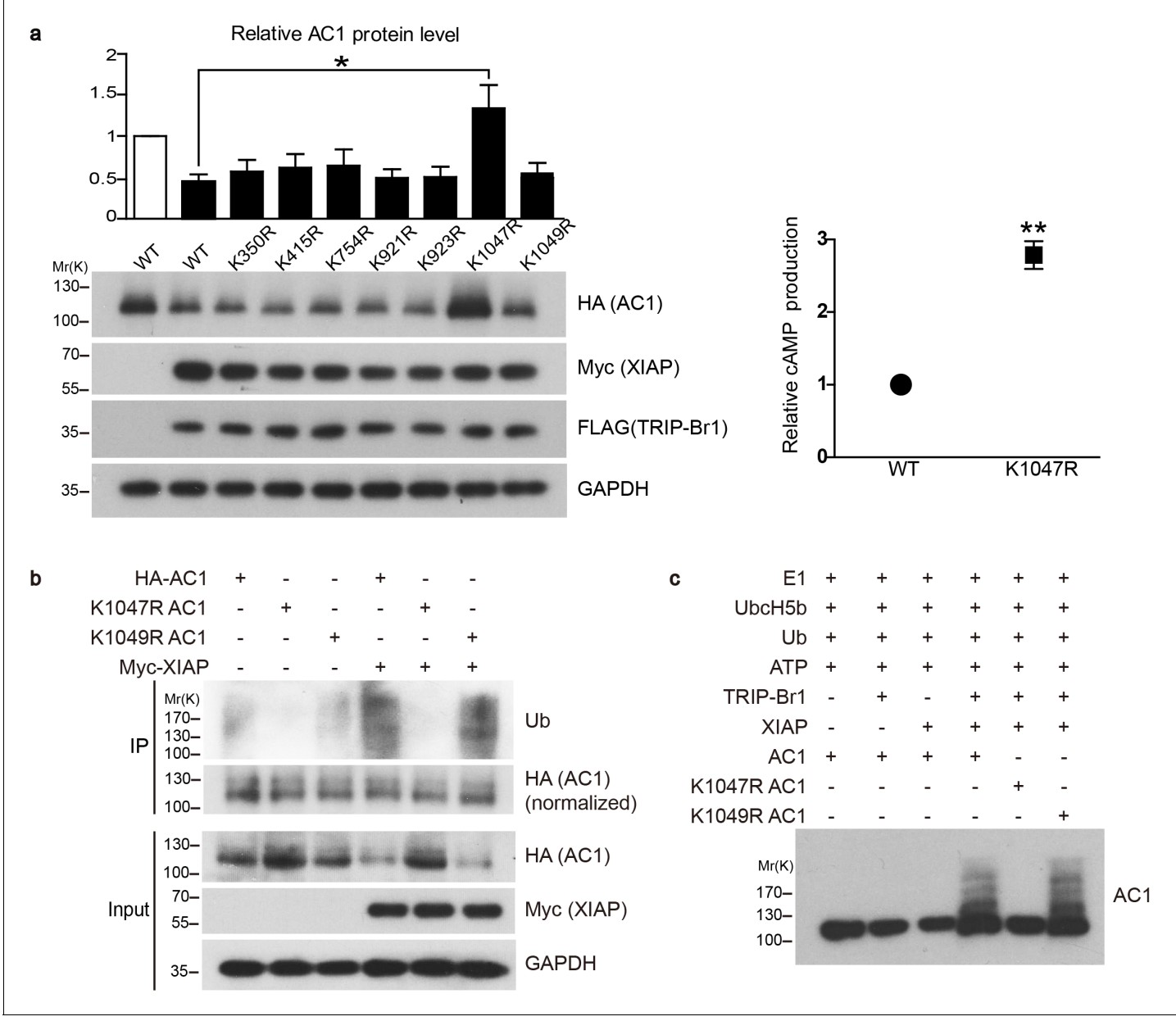

**Figure 5.** Conserved residue K1047 is the XIAP ubiquitin-conjugation site in AC1. (**a**) Left panel: Effect of Lys-to-Arg mutations in HA-tagged AC1 (HA-AC1) on XIAP-induced AC1 degradation in HEK293T cells. Quantification of the western blots is shown at the top: *, different from wild-type (WT) AC1 in the presence of XIAP, p=0.013, n = 3–6 independent experiments for various AC1 mutants. Right panel: relative cAMP production of WT and K1047R AC1 in the presence of XIAP. **p=0.0038. (**b**) Effect of K1047R and K1049R mutations in HA-AC1 on XIAP-induced ubiquitination and degradation of HA-AC1 in HEK293T cells. (**c**) In vitro ubiquitination of AC1 and its K1047R and K1049R mutants. E1, ubiquitin-activating enzyme; Ub, ubiquitin. All experiments shown here are representative of 3 independent experiments.

The following figure supplements are available for figure 5:

**Figure supplement 1.** XIAP induces K27-linked polyubiquitination of AC1.

**Figure supplement 2.** Endocytosis and recycling of HA-tagged wild-type and K1047R AC1 expressed in HeLa cells.

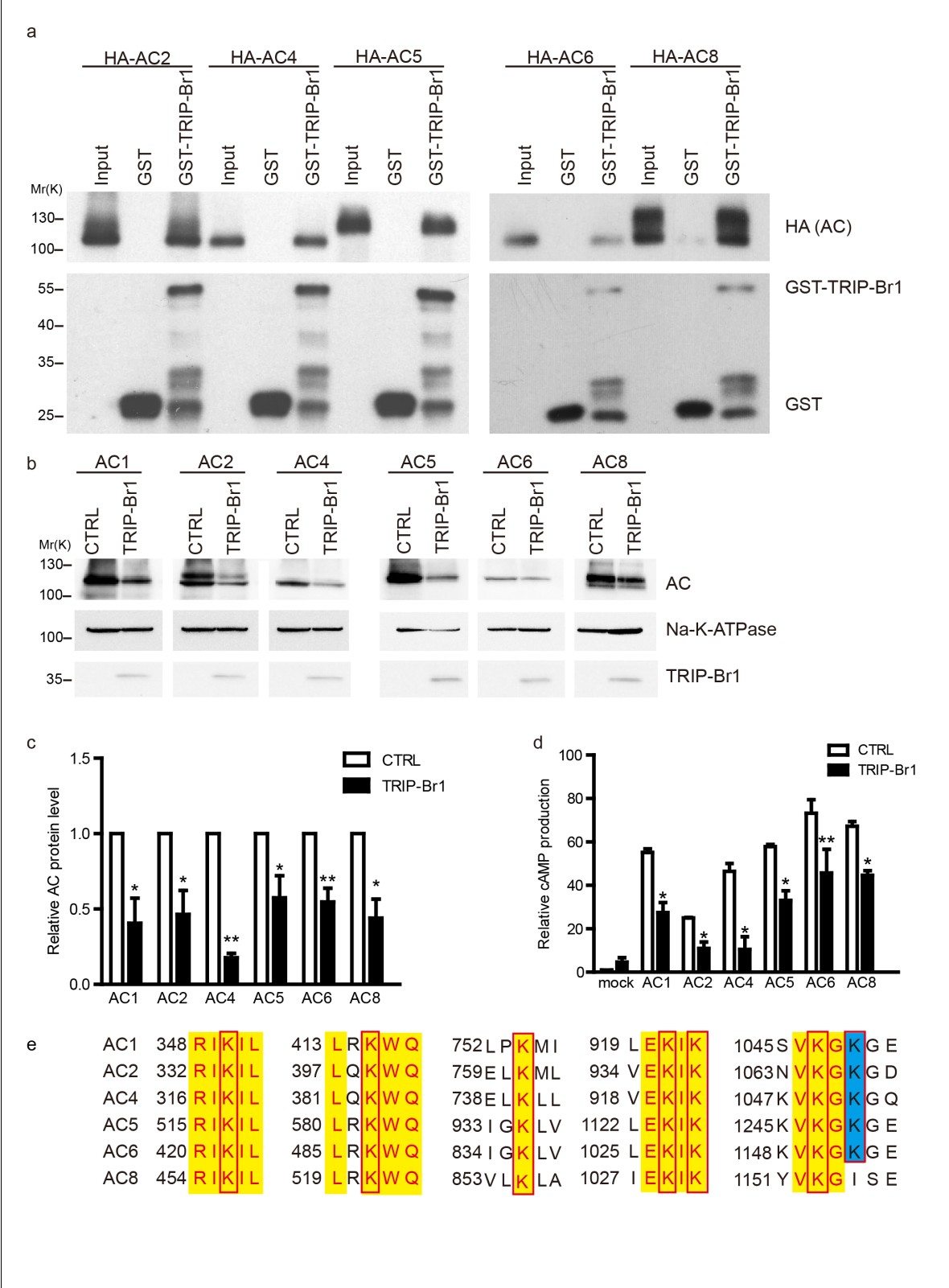

**Figure 6.** TRIP-Br1 binds to multiple AC isoforms and reduces their protein level and cAMP production. (a) GST-TRIP-Br1 captured ACs 2, 4, 5, 6, and 8.
(b) Stable expression of TRIP-Br1-V5 reduced the protein expression of ACs 1, 2, 4, 5, 6, and 8 that were transiently expressed in HEK293T cells. (c–d)
Quantification of protein levels of various AC isoforms in panel b and their forskolin-stimulated cAMP production. cAMP production is normalized
*Figure 6 continued on next page*

*Figure 6 continued*

relative to that of the control of mock cells. *, different from control, p≤0.048; **p≤0.009; ***p=0.001; n = 3–5. (e) Sequence alignment of the AC isoforms used in panel b to identify conserved Lys residues (boxed). Yellow highlight: identical residues; blue highlight: highly conserved.

suggest that TRIP-Br1 acts as an adaptor for the ubiquitination and degradation of AC1, AC2, and AC5/6—and probably also other AC isoforms—by XIAP E3 ligase (Figure 9).

## Functional changes of ACs in TRIP-Br1 knockout and XIAP ΔRING mice

The increased protein expression of multiple AC isoforms, including major cardiac isoform AC5/6 (*Sadana and Dessauer, 2009*), in the heart tissue of TRIP-Br1 knockout and XIAP ΔRING mice (*Figure 7d–e*) promoted us to examine functional changes of ACs in these two mouse strains. The basal or forskolin-stimulated cAMP production of the heart tissue of the two mouse strains was significantly higher than that of wild-type mice (*Figure 7f*); these increases were independent of phosphodiesterases (PDEs) because PDE activities were blocked by the presence of the non-selective PDE inhibitor IBMX in the cAMP assay (*Figure 7f*). Consistent with the functional increase of cardiac ACs, the heart rates, which is regulated by AC (particularly AC5/6)-cAMP-PKA signaling (*Sadana and Dessauer, 2009*), were increased by ~8–17% under basal conditions in these two mouse strains, relative to that in wild-type mice (*Figure 7g*). This difference seemed to be greater when the mice were under deeper anesthesia (~29% basal, and ~61% isoproterenol-stimulated; *Figure 7—figure supplement 3*), and the underlying mechanism is unclear at this time.

Alternatively, we assessed functional changes of ACs in the brain in TRIP-Br1 knockout mice because of increased protein expression of AC1 (*Figure 2c*) and AC 2 (*Figure 7—figure supplement 2*), two major brain AC isoforms (*Sadana and Dessauer, 2009*), and possibly other AC isoforms (*Figures 6–7*). A modest yet significant increase in CREB (cAMP response element-binding protein) phosphorylation, a classical downstream effect of AC-cAMP-PKA signaling, was observed in the hippocampus in TRIP-Br1 knockout mice (*Figure 7—figure supplement 4a–b*). We further tested the functional impact of CREB phosphorylation in the hippocampus by evaluating the mood, especially despair-like behavior, of the mice because CREB phosphorylation in certain brain regions including the hippocampus is associated with depressive and anxiety-like behaviors (*Carlezon et al., 2005*). Significantly, TRIP-Br1 knockout mice showed markedly decreased immobility time in the forced swimming test, very poor scores in the nesting test (especially in the first 24 hr), increased time in the open arms in an elevated plus maze, and moderately elongated stay in the light box in the light-dark transition test (*Figure 7—figure supplement 4e–h*), although they displayed apparently normal physical strengths as indicated by normal body weights and performance in the wire hanging test (*Figure 7—figure supplement 4c–d*); all these results suggest mood disorders, especially despair-like behaviors. The TRIP-Br1 knockout mice were also tested in the marble burying test but showed normal behavior (*Figure 7—figure supplement 4i*).

In sum, the increase in AC protein expression resulted in consistent changes in the heart and brain functions in TRIP-Br1 knockout and XIAP ΔRING mice.

## Negative-feedback regulation of AC1 by XIAP/TRIP-Br1

What is the physiological significance of XIAP/TRIP-Br1 mediated downregulation of ACs? Because AC-cAMP-PKA signaling pathway is known to stabilize XIAP through CREB or PI3K signaling (*Cho et al., 2011*; *Dan et al., 2004*; *Gao et al., 2008*; *Misra and Pizzo, 2005*), we hypothesized that an increase in AC and PKA activities upregulates XIAP protein expression and subsequently downregulates AC protein expression in a negative-feedback manner. We tested this hypothesis in mouse primary cells, specifically in macrophages, because they reportedly upregulated XIAP upon forskolin stimulation (*Misra and Pizzo, 2005*) and are relatively easy to culture. 6 hr treatment of either isoproterenol or forskolin, two commonly used AC/cAMP agonists, increased XIAP protein expression, and, more importantly, markedly decreased AC1 protein expression in the macrophages of wild-type mice in a PKA-inhibitor-sensitive manner (*Figure 8a–b*). By contrast, the two agonists had no impact on TRIP-Br1 protein expression (*Figure 8a–b*). Significantly, ablation of TRIP-Br1 gene nullified the regulatory effects of the cAMP agonists on AC1 protein expression while had little

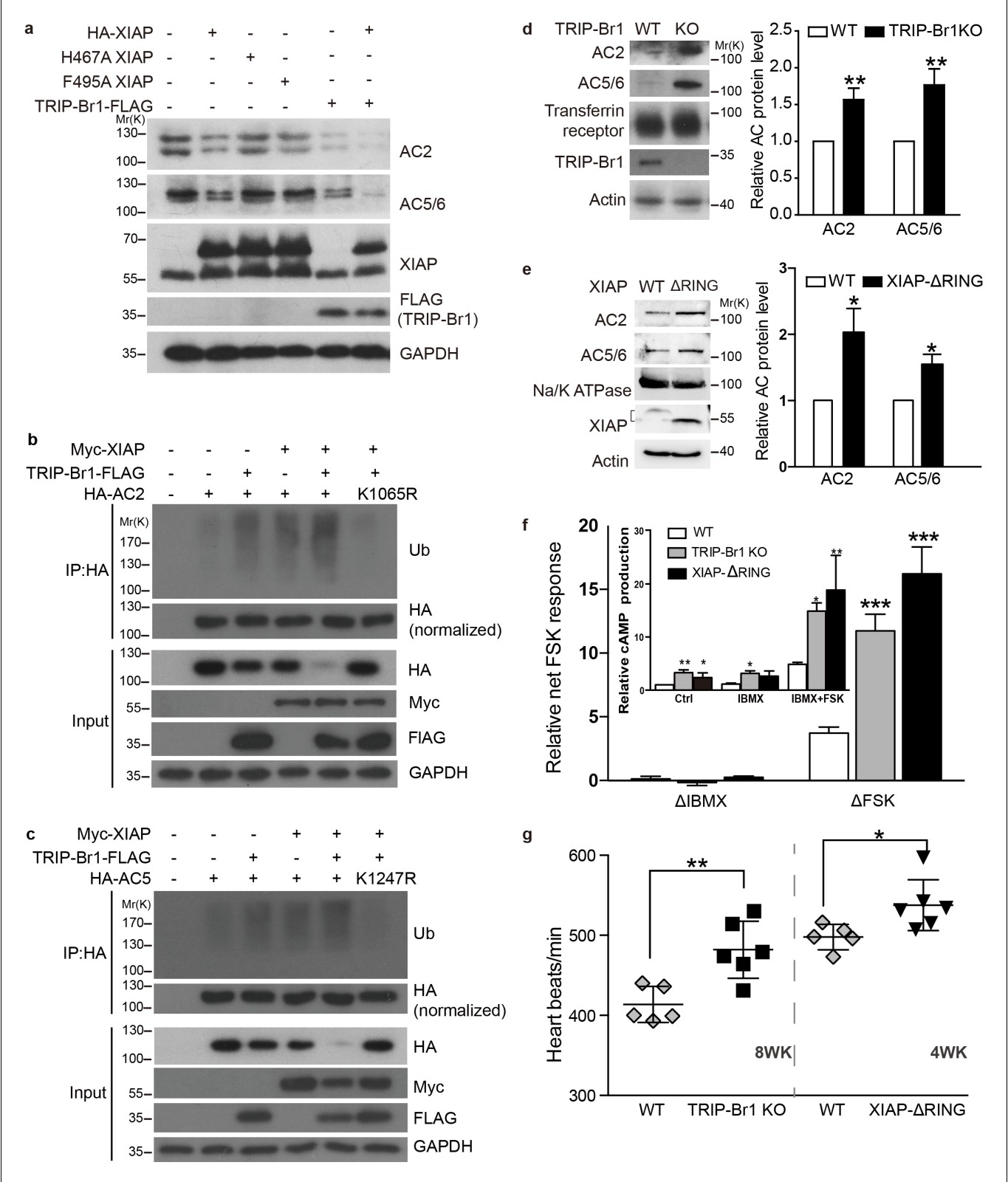

**Figure 7.** XIAP/TRIP-Br1 attenuates the expression of multiple AC isoforms. (a) Total endogenous AC2 and AC5/6 in HeLa cells were examined following overexpression of HA-XIAP and its H467A and F495A mutants and TRIP-Br1-FLAG (quantification in *Figure 7—figure supplement 1*). (b–c,) Mutation to Arg of K1065 in HA-AC2 and K1247 in HA-AC5 abolished XIAP/TRIP-Br1-induced ubiquitination and degradation of HA-AC2 (b) and HA-AC5 (c), respectively, in HEK293T cells. The experiments shown here are representative of 3 independent experiments. (d–e) Knocking out TRIP-Br1 or

*Figure 7 continued on next page*

*Figure 7 continued*

the RING domain of XIAP in mice increased the expression of AC2 and AC5/6 in heart tissue. Transferrin receptor or Na/K ATPase: membrane-protein negative control; actin: loading control. *, different for wild-type (WT), p≤0.036, n = 3; **p≤0.006, n = 8. (f), Net cAMP production in response to IBMX (ΔIBMX, 100 μM) or forskolin (ΔFSK, 10 μM) in heart tissues of wild-type (WT), TRIP-Br1 knockout, and XIAP ΔRING mice. ΔIBMX and ΔFSK are the differences between IBMX-stimulated and control cAMP production, and between forskolin/IBMX- and IBMX-stimulated cAMP production, respectively (inset). *, different from WT, p≤0.04; **p≤0.006; ***p≤0.001, n = 8 wk-old females for all three groups. (g) Basal heart rates of wild-type (WT), TRIP-Br1 knockout, and XIAP ΔRING mice. **, different from WT, p=0.0049; *, p=0.032; 8-wk-old male and female littermates of WT (n = 5) and TRI-BR1 knockout mice (n = 6) or 4-wk-old male littermates of WT (n = 5) and XIAP ΔRING mice (n = 6) were used for experiments. The mice were anesthetized with Avertin. The difference in the basal heart rates of 4-wk-old and 8-wk-old WT mice has been reported before (*Moghtadaei et al., 2016*).

The following figure supplements are available for figure 7:

**Figure supplement 1.** TRIP-Br1 bridges the interaction of AC2 and AC5/6 with XIAP E3 ligase.

**Figure supplement 2.** Knocking out TRIP-Br1 in mice increased AC2 expression in the brain.

**Figure supplement 3.** Change in the heart rate of TRIP-Br-1 knockout mice under deep anesthesia.

**Figure supplement 4.** Knocking out TRIP-Br1 induces the phosphorylation of CREB in the hippocampus and mood disorder (especially despair-like behavior) in mice.

impact on the agonists' effects on XIAP protein expression (*Figure 8c–d*). These observations indicated that cAMP-PKA signaling regulates AC1 expression through increasing XIAP in a negative-feedback manner and TRIP-Br1 is an indispensable part of this negative-feedback loop.

## Discussion

### A general mechanism for ubiquitinating and degrading tmACs

Ubiquitination regulates the trafficking and degradation of several molecules in the G-protein-coupled receptor (GPCR) signaling pathway, such as GPCRs, Gαs, beta-arrestin1/2, GRK2, RGS7, PDE4D5, and PKA (*Wojcikiewicz, 2004*; *Xiang, 2011*; *Lignitto et al., 2011*; *Naviglio et al., 2004*), which are either critical upstream modulators or downstream targets of cellular AC functions. Thus, it is surprising that almost nothing is known about how the degradation of AC per se is regulated through ubiquitination or other mechanisms. Two early studies suggested that AC degradation was mediated by the lysosome (*Revis and Durham, 1979*) or the proteasome (*Tang et al., 2008*), but the mechanisms were unclear; and, PAM, an E3 ubiquitin ligase, was reported to bind and inhibit AC5 and AC6, although its inhibitory effect on the ACs was apparently independent of the ubiquitin-ligase activity (*Pierre et al., 2004*). Thus, our findings provide the first mechanism of AC degradation, and, notably, this mechanism appears to apply to 6 out of 9, if not all, tmAC isoforms (*Figure 9*).

Distinct AC isoforms share high structural homology, particularly in their 2 catalytic domains, and thus it is not unexpected that AC isoforms are controlled by common regulatory mechanisms. For instance, Gsα and P-site analogs bind to all tmACs and modulate their activity, and forskolin binds and stimulates all tmACs except AC9. Our study adds another common regulatory mechanism to this list. As in the case of other regulatory proteins and chemicals that control tmACs, the highly conserved C1b and C2 domains of tmAC contain the binding site of TRIP-Br1 and the ubiquitination site of XIAP, respectively (*Figure 9*).

### TRIP-Br1 as an adaptor of E3 ligase

TRIP-Br1 and other members of its family contain an N-terminal putative cyclin-A-binding domain, a novel highly conserved SERTA (Sei-I, RBT1, and TARA) domain of unclear function, a binding motif for PHD zinc finger- and/or bromodomain-containing proteins, and an acidic C-terminal domain (*Lai et al., 2007*; *Hsu et al., 2001*), and these multiple protein-interaction domains in TRIP-Br1 enable it to act as an adaptor/scaffold protein and tethers different signaling components (*Hsu et al., 2001*; *Sim et al., 2006*). Moreover, TRIP-Br1 appears to modulate their activity and

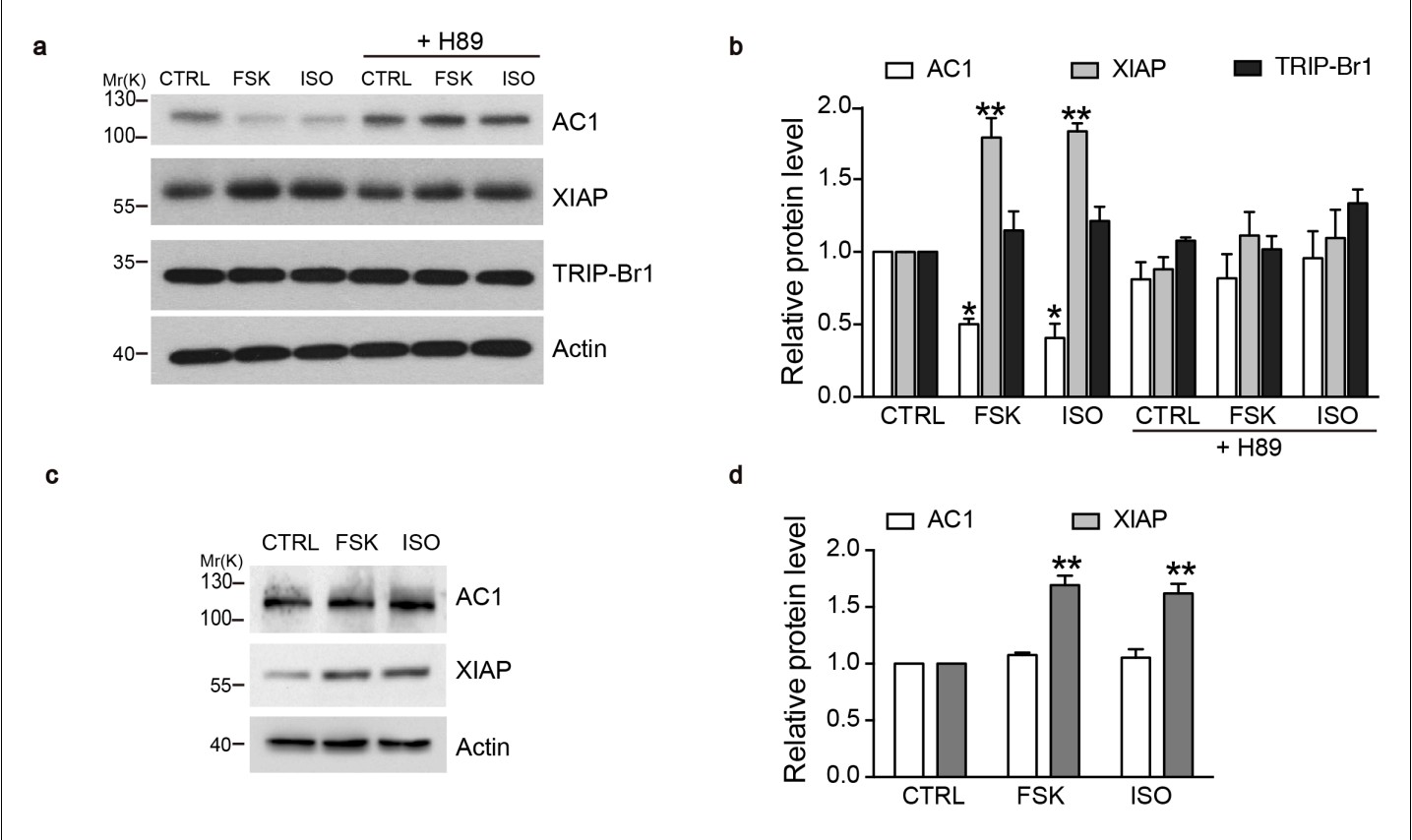

**Figure 8.** cAMP agonists promote XIAP upregulation and AC1 degradation in a PKA-dependent manner in the macrophages of wild-type but not TRIP-Br1 knockout mice. Changes in AC1, XIAP, and TRIP-Br1 expression with or without 6 hr forskolin (FSK, 10 µM, in the presence of 100 µM IBMX) or isoproterenol (ISO,10 µM) treatment in the microphages of wild-type (**a–b**) and TRIP-Br1 knockout (**c–d**) mice. H89 (10 µM), PKA inhibitor; β-actin, loading control. (**b and d**), Summary data of panels **a** and **b**, respectively, relative to β-actin. Different from control (CTRL), *p≤0.0427; **p≤0.0076, n = 3.

expression of some of its interacting proteins (*Sugimoto et al., 1999*; *Hong et al., 2011*; *Lee et al., 2009*); however, whether this function of TRIP-Br1 needs a third protein is unclear. This study has shown that TRIP-Br1 acts as an adaptor by tethering ACs on its SERTA domain and XIAP E3 ligase on the N-terminal domain (which includes a putative cyclin-A-binding domain). The SERTA domain has been suggested to be involved in CDK4 binding and possibly homo-/hetero-dimerization (*Hsu et al., 2001*); by contrast, the function of the N-terminal putative cyclin-A-binding domain is largely unknown (*Hsu et al., 2001*). Our study has revealed previously unrecognized functions of these domains.

The adaptor proteins of E3 ubiquitin ligases not only facilitate specific interactions between E3 ubiquitin ligases and their substrates: they also alter the subcellular localization, activity, and stability of the corresponding ubiquitin ligases (*Léon and Haguenauer-Tsapis, 2009*). For example, the transcriptional regulator Smad7 binds to Smurf1/2 (2 Nedd4-family E3 ligases), and the nucleus-to-cytoplasm translocation of the Smad7-Smurf1/2 complex is triggered by TGF-β receptor activation (*Léon and Haguenauer-Tsapis, 2009*; *d'Azzo et al., 2005*). Smad7 not only acts as an adaptor for mediating the ubiquitination of TGF-β receptors by Smurf1/2, but also increases the activity of Smurf1/2 in several ways, including by blocking the auto-ubiquitination and degradation of Smurf1/2 (*Léon and Haguenauer-Tsapis, 2009*; *d'Azzo et al., 2005*). This is reminiscent of TRIP-Br1: like Smad7, TRIP-Br1 is a multi-functional protein, including regulating transcription and acting as an adaptor for XIAP E3 ligase. It is noted that cIAP1 and cIAP2, two IAP proteins highly homologous to

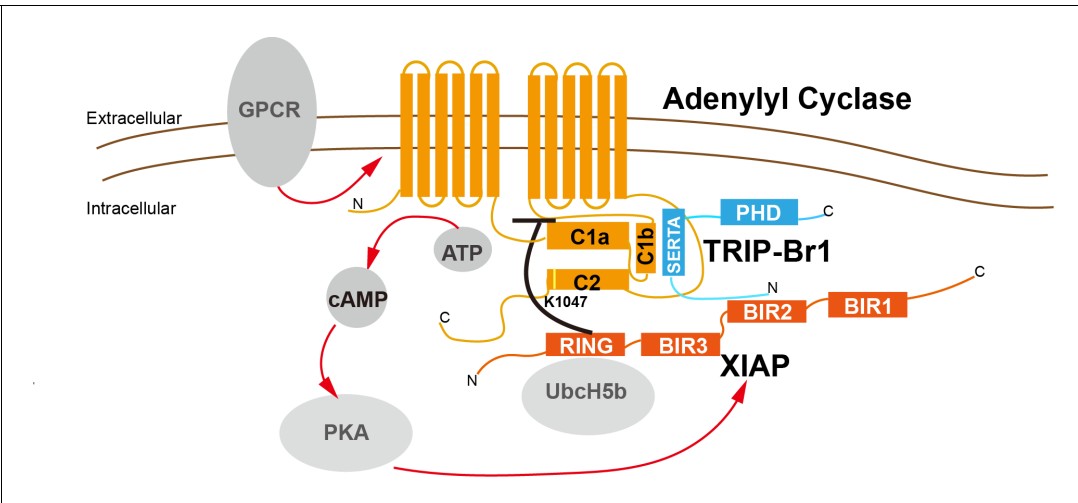

**Figure 9.** A model depicting a general negative-feedback mechanism of ubiquitination and degradation of multiple AC isoforms. The SERTA domain of TRIP-Br1 binds to the C1b catalytic domain of AC1 or other AC isoforms and subsequently recruits XIAP, which, together with the E2 enzyme UbcH5b (or UbcH5a or UbcH5c) (*Nakatani et al., 2013*; *Mace et al., 2008*), ubiquitinates K1047 in AC1 or an equivalent Lys residue in other AC isoforms. The N-terminus of TRIP-Br1 interacts with XIAP, probably with its BIR2 domain (*Hong et al., 2009*), and the Ring domain of XIAP interacts with UbcH5b (*Nakatani et al., 2013*; *Mace et al., 2008*). The ubiquitination of ACs leads to their endocytosis and degradation. Sustained AC activation and cAMP production under catecholamine stress increase XIAP protein expression through PKA, and elevated XIAP protein expression subsequently dstabilizes ACs and eventually lows cAMP production/signaling. Red arrow, stimulatory effect; black dash: inhibitory effect.

XIAP, had virtually no functional interaction with TRIP-Br1/AC1 complex (*Figure 4—figure supplement 2*), suggesting that XIAP specifically targets ACs.

An early study showed TRIP-Br1 knockout mice displayed no overt pathology, except for certain abnormalities in pancreatic islets, including a reduction in the number of islets and the area of beta-cells, impaired insulin secretion, and glucose intolerance; these presumably resulted from the loss of TRIP-Br1's regulatory function in cell cycle (*Fernandez-Marcos et al., 2010*). We observed a large increase in AC protein expression and cAMP production in the heart of TRIP-Br1 knockout mice, and this was accompanied by elevation in resting and stimulated heart rates (*Figures 2c* and *7d–g*, and *Figure 7—figure supplements 2–3*). Pathological effects of a marked increase in resting and stimulated heart rates might not be manifested in mice that are housed in a stress-free environment in the laboratory, at least not within the short period for which we observed the animals (*Sadana and Dessauer, 2009*). Similar results on the heart rate were obtained with mice that overexpressed AC5/AC6 (*Sadana and Dessauer, 2009*). Overexpression of ACs has also been previously shown to result in functional changes in the brain, such as alcohol dependency, learning and memory, and synaptic plasticity (*Sadana and Dessauer, 2009*; *Garelick et al., 2009*), and long-term studies are required to analyze these potential changes in TRIP-Br1 knockout mice. Interestingly, accompanying a robust increase in AC protein expression in the brain of (*Figure 2* and *Figure 7—figure supplement 2*), we found a modest yet significant increase in CREB phosphorylation in the hippocampus and conspicuous mood disorders, especially despair-like behavior, in TRIP-Br1 knockout mice (*Figure 7—figure supplement 4*).

The activity and expression of TRIP-Br1 are altered in several physiological and pathological processes. TRIP-Br1 activity is modulated through its dephosphorylation by physically associated PP2A, but the kinase(s) that phosphorylates TRIP-Br1 remains unidentified (*Zang et al., 2009*). Furthermore, TRIP-Br1 is induced in neurons following nerve growth factor deprivation and β-amyloid treatment and is required for neuronal cell death (*Biswas et al., 2010*), and TRIP-Br1 expression is also elevated in a cellular model of Parkinson's disease (*Ryu et al., 2005*) and in cancer cells (*Hong et al., 2009*). Because cAMP promotes neuronal survival (*Li et al., 2000*) and suppresses Parkinson's disease (*Muda et al., 2014*) and tumor formation (*Bierie and Moses, 2006*), future studies that enable segregation of multiple function of TRIP-BR1 must further investigate how and to what

extent the regulatory function of TRIP-Br1 in AC-cAMP signaling, in addition to its transcriptional function, contributes to neuronal and other deficiencies.

## XIAP E3 ligase and negative-feedback regulation of AC-cAMP-PKA signaling

The E3 ubiquitin-ligase function of XIAP is less well understood as compared to its inhibitory effect on caspases (*Galbán and Duckett, 2010*). The identified substrates of XIAP E3 ligase include XIAP itself, XIAP-interacting proteins involved in apoptosis, and target proteins that perform other physiological functions, such as TAK1, MEKK2/3, and PTEN (*Galbán and Duckett, 2010*; *Van Themsche et al., 2009*; *Takeda et al., 2014*). In addition, XIAP, independently of its E3 ligase, promotes the ubiquitination of C-RAF by recruiting CHIP ubiquitin ligase (*Dogan et al., 2008*). Although XIAP ΔRING mice, similar to XIAP knockout mice, are viable and apparently healthy, they showed increased expression of mutant XIAP (*Schile et al., 2008*) (*Figure 4d*) and unexpected increased caspase activity (*Schile et al., 2008*). These data indicate that XIAP E3 ligase activity targets XIAP itself and caspases. Our data demonstrated that XIAP ΔRING mice had increased AC expression and cAMP production (*Figures 4d* and *7e–f*), revealing that ACs are previously unidentified targets of XIAP E3 ligase activity in vivo.

XIAP is upregulated by AC-cAMP-PKA signaling through either PI3K signaling or CREB phosphorylation (*Cho et al., 2011*; *Dan et al., 2004*; *Gao et al., 2008*; *Misra and Pizzo, 2005*). Our data suggested that stimulating ACs through the AC agonists isoproterenol and forskolin increased XIAP protein expression and decreased AC1 protein expression in a PKA-dependent manner in mouse macrophages (*Figure 8*). Because XIAP destabilizes multiple AC isofroms, this negative-feedback regulation could apply to multiple AC isoforms and provide a mechanism to control the homeostasis of cAMP signaling (*Figure 9*). Various types of acute and chronic stressors trigger the systematic release of stress hormones including catecholamines that activates AC-cAMP-PKA signaling pathway, and acute and chronic catecholamine stress impairs cardiac myocytes (*Okumura et al., 2007*), wound healing (*Kim et al., 2014*), and innate immune defense (*Sandrini et al., 2010*). TRIP-Br1/XIAP-mediated negative-feedback regulation (*Figure 9*) could protect the body from such detrimental effects induced by catecholamine stress.

XIAP also interacts with TGF receptors and associated proteins (*Galbán and Duckett, 2010*). Notably, all 3 TGF-$\beta$ isoforms upregulate XIAP at the transcriptional level and thereby decrease PTEN expression (*Van Themsche et al., 2007*, *2010*). Our findings raise the possibility that TGF-$\beta$ downregulates ACs by enhancing XIAP expression. Intriguingly, treatment with TGF-$\beta$1 lowered AC activity in cardiomyocytes and prostate cancer cells, which suggests a possible reduction of AC protein expression (*Steiner et al., 1994*; *Nair et al., 1995*). However, our preliminary study did not consistently observe TGF-dependent change of AC1 in mouse macrophages (not shown). In addition to TGF-$\beta$, XIAP is reportedly regulated by several other signaling pathways (*Hiramatsu et al., 2014*; *Hatton et al., 2011*). Future work is needed to understand whether these signaling pathways are involved in the regulation of ACs, particularly in specific cell types.

In summary, ACs are critically important signaling molecules that are involved in diverse physiological processes such as cell growth and differentiation, learning and memory, olfaction, cardiac stress, and epithelial secretion. This study advances our understanding of the regulation of tmACs through protein-degradation mechanisms and identifies a previously unrecognized link between cAMP signaling and numerous signaling pathways that regulate either XIAP or TRIP-Br1. Particularly, we found that XIAP/TRIP-Br1-mediated degradation of tmACs forms part of a negative-feedback loop that controls the homeostasis of cAMP signaling. Our study can potentially open up new avenues of research and development for designing therapeutic approaches to control AC functions in health and disease.

## Materials and methods

### Antibodies

Antibodies against the following proteins and tags were from commercial sources: TRIP-Br1, Enzo Life Science (RRID:AB_2052741); XIAP, R and D Systems (RRID:AB_2215008); AC1 (RRID:AB_2223098), AC5/6 (RRID:AB_2257941), ubiquitin (RRID:AB_778730), GAPDH (RRID:AB_10847862),

and $\beta$-actin (RRID:AB_626632), Santa Cruz Biotechnology; AC2, Novus Biologicals; V5 (RRID:AB_2556564) and transferrin receptor (RRID: AB_86623), Life Technologies; HA (RRID:AB_2314672), Covance; Ubiquitin (linkage-specific K27, K48, and K63), Abcam; Phospho-CREB, Cell Signaling Technology; NeuN, Millipore; and Na-K-ATPase ($\alpha$ subunit) (RRID:AB_258029) and FLAG M2 (antibody RRID:AB_439685 and affinity gel RRID:AB_10704031), Sigma-Aldrich.

## cDNAs and plasmids

The cDNAs of human TRIP-Br1 were amplified from Calu-3 cells by means of reverse transcription-PCR performed using SuperScript Reverse Transcriptase with oligo(dT)$_{12-18}$ and Platinum Taq DNA polymerase (Invitrogen). The cDNAs of membrane ACs were kind gifts from other researchers: bovine AC1, Dr. Daniel R. Storm (University of Washington); rat AC2, Dr. Randall Reed (Johns Hopkins University); and rat AC4, rabbit AC5, rat AC6, and rat AC8, Dr. Zvi Vogel (Weizmann Institute of Science). The cDNAs of the membrane ACs were PCR-amplified and cloned into pCMV-3HA vectors. pRK5-Myc-XIAP was provided by Dr. Kenny Chung (Hong Kong University of Science and Technology), and pcDNA3-HA-XIAP and pcDNA3-HA-XIAP F495A were kind gifts from Dr. Philipp Jost (Technische Universität München). All XIAP cDNAs encoded human XIAP.

## Cell culture and transfection

HEK293T (RRID: CVCL_1926) and HeLa (RRID: CVCL_0030) cells were obtained from ATCC. These cells were assumedly authenticated by ATCC and were not further authenticated in this study. All cell lines were routinely tested negative for mycoplasma contamination and maintained in Dulbecco's Modified Eagle Medium (DMEM) supplemented with 10% fetal bovine serum and 100 U/mL penicillin/streptomycin (Life Technologies) in an atmosphere of 95% air-5% $CO_2$ at 37°C. All transfections were performed using the Lipofectamine 2000 reagent kit (Invitrogen).

## Mice

TRIP-Br1(encoded by *Sertad1* gene) knockout mice (RRID: MGI:4437096) in a C57BL/6 genetic background were a kind gift from Dr. Manuel Serrano (Spanish National Cancer Research Center). XIAP *ΔRING* mice (RRID: MGI:3809793) was made by deleting the ring domain from the *Xiap* gene in a C57BL/6 genetic background as previously described (*Schile et al., 2008*). Both mouse strains were genotyped using a PCR assay, according to previous reports (*Schile et al., 2008*; *Fernandez-Marcos et al., 2010*). Unless indicated otherwise, wild-type mice were strain-, sex-, and age-matched with TRIP-Br1 knockout or XIAP *ΔRING* mice. All animal procedures were approved by the University Committee on Research Practices at the Hong Kong University of Science and Technology (the ethics protocol number 2014028).

## Co-immunoprecipitation (co-IP)

Co-IPs were performed using procedures described previously (*Sun et al., 2012*), with minor modifications. After incubation with whole-cell extracts, Protein A Sepharose beads were washed 3 times with 500 µL of a 'Co-IP' lysis buffer (*Sun et al., 2012*) plus 0.5% Triton X-100.

## Pull-down assays

GST or GST fusion proteins (5 µg) were incubated with glutathione-Sepharose beads (GE Healthcare) for 2 hr at 4°C in the Co-IP lysis buffer (*Sun et al., 2012*), and then centrifuged at 500 × *g* for 1 min to pellet the beads; the beads were washed once with 500 µL of the Co-IP buffer. HEK293T cells transfected with various expression vectors for 24 hr were lysed with the Co-IP buffer (500 µL for cells in one 60 mm culture dish) and then centrifuged at 16,000 × *g* for 20 min. Next, the obtained supernatants were incubated (overnight, 4°C) with the Sepharose beads bound with GST/GST fusion proteins, after which the beads were washed five times (500 µL each time) with the Co-IP buffer plus 0.5% Triton X-100. The proteins bound on the beads were eluted using 60 µL of 1× SDS-PAGE loading buffer and analyzed by means of western blotting.

## Pairwise pull-down assays

For pairwise pull-down assays, all proteins were expressed in and purified from *E. coli* strain BL21 codon plus. The anti-FLAG M2 affinity gel (Sigma) was washed twice with a washing buffer (50 mM

Tris-HCl, 150 mM NaCl, pH 7.4) and resuspended in the Co-IP buffer to prepare a 50% slurry, and then 5 µg of each protein and 40 µL of the gel suspension were mixed with the Co-IP buffer in a final volume of 500 µL and incubated for 2 hr at 4°C. Next, the gel was washed thrice with 500 µL of phosphate-buffered saline (PBS) containing 1% Triton X-100, and the bound proteins were eluted by incubating the resin with 100 µL of 0.1 M glycine-HCl, pH 3.5, for 5 min at room temperature. The resins were centrifuged for 30 s at $5000 \times g$, and then the supernatants were transferred to fresh test tubes containing 10 µL of 0.5 M Tris-HCl, 1.5 M NaCl, and pH 7.4; lastly, $6\times$ SDS-PAGE loading buffer was added and the samples were subjected to western blotting analysis.

## Preparation of mouse tissues for western blotting

Tissues were dissected from mice, washed with PBS, and homogenized in RIPA buffer (50 mM Tris-HCl, 150 mM NaCl, 0.1% SDS, 0.5% sodium deoxycholate, 1% Triton-100, 1 mM DTT, 1 mM PMSF, pH 7.4) supplemented with $1\times$ protease-inhibitor cocktail; 500 µL of RIPA buffer was used for 30 mg of tissues. After sonication, tissue lysates were centrifuged at $16,000 \times g$ for 20 min and the supernatants were used for western blotting.

## Cycloheximide experiments

HeLa cells with or without TRIP-Br1 shRNA treatment were treated with 200 µM CHX (Sigma) for various times. Subsequently, cells in both sets of experiments were lysed in radioimmunoprecipitation assay (RIPA) buffer (150 mM NaCl, 1.0% NP40, 0.5% sodium deoxycholate, 0.1% SDS, 50 mM Tris, pH 7.4) supplemented with a $1\times$ protease-inhibitor cocktail (Roche). Cell lysates were analyzed by performing western blotting.

## RNA interference

Two different TRIP-Br1, and three different XIAP shRNAs (*Van Themsche et al., 2009*), and a scrambled and non-targeting control sequence were cloned into pSuper basic vectors. The sequences of the shRNAs are listed below:

shTRIP-Br1-1: 5′-gggcctgtttgaggatatt-3′;
shTRIP-Br1-2: 5′-GACACCTCTATGTATGACAAT-3′;
shXIAP-1: 5′-GCCACGCAGTCTACAAATTCT-3′;
shXIAP-2: 5′-AGCTGTAGATAGATGGCAATA-3′;
shXIAP-3:5′-GCACTCCAACTTCTAATCAAA-3′;
sh-scrambled: 5′-ACGCATGCATGCTTGCTTT-3′.

Plasmids were transfected into HeLa cells by using Lipofectamine 2000 according to manufacturer instructions, and 24 hr after transfection, the cells were lysed with the Co-IP buffer supplemented with $1\times$ protease-inhibitor cocktail (Roche) and 20 mM N-ethylmaleimide (NEM). Lysates were used for either western blotting or AC1 immunoprecipitation for ubiquitination assays.

## Site-directed mutagenesis

Point mutations in ACs were generated using a modified QuikChange reaction. KAPA HIFI polymerase PCR mixtures containing QuikChange primers and 20 ng of template were amplified for 16 cycles, and the PCR products were resolved using agarose gels and purified using a gel purification kit (Viogene). The purified DNA was digested with Dpn1 for 1 hr at 37°C, and this was followed by transformation into the bacterial strain DH5α. Clones were sequenced and plasmids were extracted using a miniprep kit (Viogene).

## In vitro and in vivo ubiquitination assays

His-AC1 was expressed in and purified from HEK293T cells. Briefly, 48 hr after transfection, cells were lysed using a Tris-NP-40 solubilization buffer: 20 mM Tris, 150 mM NaCl, 2 mM $MgCl_2$, 10% glycerol, 5 mM imidazole, 0.1% NP-40, protease-inhibitor cocktail (Roche), 1 mM DTT, 1 mM PMSF, 5 mM ATP, pH 7.4. After sonication, intact cells and cell debris were removed by centrifuging the lysates at $16,000 \times g$ for 20 min at 4°C. The supernatants were filtered through a porous membrane (pore diameter, 0.22 µm) and incubated with Ni-NTA beads (GE Healthcare) for 4 hr at 4°C. Subsequently, the beads were washed thrice (10 mL each time) with a washing buffer (same composition as the solubilization buffer, but containing 500 mM NaCl and 3 mM ATP) and eluted using an elution

buffer (20 mM Tris, 150 mM NaCl, 2 mM MgCl₂, 10% glycerol, 400 mM imidazole, 0.1% NP-40, pH 7.4). The eluted proteins were transferred to a ubiquitination reaction buffer (25 mM Tris-HCl, 5 mM MgCl₂, 100 mM NaCl, 1 mM DTT, 2 mM ATP, pH 7.6) and concentrated to 1 mg/mL by using Amicon Ultra-15 Centrifugal Filter Units (Millipore; MWCO, 10 kD).

In vitro ubiquitination assays were performed following previously described procedures (*Okiyoneda et al., 2010*; *Liu et al., 2014*) and using an in vitro ubiquitination kit (ENZO Life). Briefly, purified His-AC1 was incubated with components of the putative ubiquitination machinery, including 100 ng of UBA1 (E1, ubiquitin-activating enzyme), 150 ng of UbcH5b (E2, ubiquitin-conjugating enzyme), 0.5 µg of XIAP, 2.5 µg of His-AC1, and 5 µg of ubiquitin, in a reaction buffer in a final volume 50 µL; the reaction buffer contained 25 mM Tris-HCl, 5 mM MgCl₂, 100 mM NaCl, 1 mM DTT, and 2 mM ATP (pH 7.6). The mixture was incubated at 30°C for 4–6 hr and the reaction was terminated by adding 1× SDS-PAGE loading buffer. Ubiquitination of His-AC1 was analyzed by immunoblotting the proteins with anti-AC1 antibodies.

For in vivo ubiquitination assays performed for checking the ubiquitination status of endogenous AC1, HeLa cells were lysed (in the Co-IP buffer) 48 hr after transfection with various plasmids. The cell lysates were centrifuged at 16,000 × *g* at 4°C for 15 min, and 20% of the collected supernatants were immunoblotted to confirm the expression of each protein; the remaining supernatants were incubated with 1 µg of anti-AC1 antibody and 20 µL of Protein G Sepharose (GE Healthcare) overnight at 4°C. Immunoprecipitation experiments were performed in the Co-IP buffer supplemented with 1× protease-inhibitor cocktail (Roche) and 20 mM NEM. The recovered beads were washed three times with the Co-IP buffer supplemented with 0.5% Triton X-100 and heated in 1× SDS-PAGE sample loading buffer at 85°C for 20 min; the eluted proteins with immunoblotted with anti-AC1 or anti-ubiquitin antibodies (Santa Cruz Biotechnology).

## Protein purification

Full-length TRIP-Br1 and its GST-fused fragments were expressed in and purified from *E. coli* strain BL21 codon plus. Briefly, GST expression vectors were transformed into *E. coli* and incubated overnight at 37°C in LB culture plates. Single colonies were picked and transferred into 20 mL of Terrific Broth (TB) and cultured overnight at 37°C. Subsequently, bacterial suspensions were transferred into 1 L of TB and cultured at 37°C for 3–4 hr (until OD600 reached 0.5–0.8), after which 0.5 mM IPTG was added and the cultures were incubated overnight at 16°C. Bacteria were collected through centrifugation (4000 × *g*, 5 min), washed once with 20 mL of PBS, and then pelleted again. Bacterial pellets were resuspended in 1× PBS and lysed using the French Pressure method. Cell debris and unbroken cells were removed through centrifugation at 12,000 × *g* for 20 min, and the collected supernatants were incubated (2 hr, 4°C) with glutathione-Sepharose beads (GE Healthcare). The beads were washed with PBS containing 1% Triton X-100 and eluted using reduced glutathione, and the eluted samples were dialyzed against 1× PBS for 24 hr.

His-UbcH5b (an E2 enzyme for XIAP [*Nakatani et al., 2013*; *Mace et al., 2008*]), His-Ub, His-TRIP-Br1, His-XIAP, His-AC1-N-FLAG, His-AC1-M-FLAG, and His-AC1-C2-FLAG were expressed in *E. coli* BL21 cells and purified using Ni-NTA beads (Qiagen). When purifying XIAP, 200 µM zinc sulfate was added to the culture medium and all solutions used for purification in order to maintain the activity of XIAP, because it is a zinc-coordinating protein (*Gyrd-Hansen et al., 2008*). Bacteria were cultured and collected using the same procedures as those described for the purification of GST proteins, and the collected bacteria were lysed in a lysis buffer (20 mM Tris-HCl, 500 mM NaCl, 10 mM imidazole, 10% glycerol, 1 mM PMSF, 1 mM DTT, pH 8.0). Next, cell debris and unbroken cells were removed through centrifugation (12,000 × *g*, 20 min), and the obtained supernatants were incubated with Ni-NTA beads (2 hr, 4°C). The beads were washed thrice (10 mL each time) with a washing buffer (20 mM Tris-HCl, 500 mM NaCl, 20 mM imidazole, 10% glycerol, 1 mM PMSF, 1 mM DTT, pH 8.0) and eluted using an elution buffer (20 mM Tris-HCl, 500 mM NaCl, 250 mM imidazole, 10% glycerol, 1 mM PMSF, 1 mM DTT, pH 8.0); the eluted samples were dialyzed (24 hr, 4°C) against a dialysis buffer (25 mM Tris-HCl, 150 mM NaCl, 10% glycerol, 1 mM DTT, pH 8.0). Protein concentrations were measured using a NanoDrop instrument and the concentration and quality of proteins were further verified using SDS-PAGE.

His-AC1-M-FLAG and His-AC1-C2-FLAG were mainly present in inclusion bodies. To purify these proteins, inclusion bodies were dissolved using 8 M urea, and the dissolved inclusion bodies were

passed through a 25G needle 20 times. Proteins were refolded by sequentially diluting the urea to 4, 2, and 1 M with the lysis buffer. Insoluble components were removed through centrifugation (12,000 *g*, 20 min), and the collected supernatants were passed through 0.22 μm filters and then incubated with Ni-NTA beads for 2 hr at 4°C; the remaining procedures were the same as those described above.

## Cell-surface biotinylation

HEK293T cells stably expressing or not expressing TRIP-Br1 were seeded into 35 mm dishes 24 hr before labeling with biotin. Surface biotinylation was performed by using the cell-surface protein biotinylation and purification kit from Pierce with sulfo-NHS-SS-biotin, a thiol-cleavable amine-reactive biotinylation reagent. Briefly, cells were washed three times with PBS containing 1 mM $MgCl_2$ and 0.1 mM $CaCl_2$ (PBS-C/M) and then 1 mL of sulfo-NHS-SS-biotin (1 mg/mL in ice-cold PBS-C/M) was added. After gentle shaking for 30 min at 4°C, the cells were washed three times with the PBS-C/M solution and incubated with a quenching buffer (50 mM glycine in PBS-C/M) to remove excess biotin. Next, the cells were washed twice with PBS-C/M, lysed with RIPA buffer supplemented with 1× protease-inhibitor cocktail (Roche), and incubated overnight at 4°C with a slurry of neutravidin agarose beads (Thermo Scientific). The captured biotinylated proteins were eluted in 1× SDS-PAGE sample loading buffer supplemented with 50 mM DTT for 2 hr at room temperature and used for western blotting.

## Measurement of heart rate in mice

Heart rates were measured using a previously described two-channel electrocardiogram (ECG) (*Zehendner et al., 2013*), with certain modifications. Briefly, mice were anesthetized by intraperitoneally injecting them with an anesthetic and then placed on a 37°C heating plate., The anesthetic was 2.5% Avertin in PBS plus 2.5% DMSO (180 μL/20 g bodyweight) in most experiments, and 2 parts 10% ketamine, 1 part 2% xylazine, and 12 parts PBS (150 μL/20 g bodyweight) in some experiments. A small incision in the skin was made in the left paw and right lower abdomen for inserting a silver wire electrode subcutaneously. The ECG was recorded by using an Axopatch 200B amplifier and Axon DigiData1322A with Axon pClamp9 software; a 16 kΩ resistor was added in series with the amplifier to reduce the amplitude of the ECG and to avoid overshoot of the amplifier. The ECG was acquired at a sampling rate of 1 kHz and filtered at 5 kHz. The heart beats per minute were manually counted from 10 s recordings.

After recording a stable resting heart rate for at least 2 min, 0.04 mg isoproterenol (in PBS)/kg bodyweight was injected into mice intraperitoneally. Subsequently, the ECG was acquired for at least 8 min. The heart rates at 4 min after isoproterenol injection were counted as the isoproterenol-induced heart rates.

## Intracellular cAMP assays

Cells transfected with ACs were washed and incubated with or without 10 μM forskolin for 8 min. After washing once with room-temperature PBS, the cells were lysed by incubating them for 20 min with 0.1 M HCl at room temperature. The cell lysates were used for measuring cAMP levels (CYCLIC AMP EIA KIT; Biomedical Technologies Inc.) and protein concentrations (DC assays; BioRad).

## Endocytosis and recycling assays

HeLa cells were transfected with various plasmids for 24 hr before experiments. Cell-surface proteins were biotinylated with the thiol-cleavable biotinylation reagent at 4°C as described above, and the internalization of biotinylated proteins was initiated by incubating cells at 37°C. Biotin was stripped from the biotinylated proteins remaining on the cell surface by using a MESNA buffer (50 mM MESNA, 100 mM NaCl, 50 mM Tris, 1 mM $MgCl_2$, 0.1 mM $CaCl_2$, pH 8.6) (1 hr, 4°C), and then the cells were either lysed in RIPA buffer for assessing the amount of internalized AC1 or again incubated for 10 min at 37°C to trigger the recycling of internalized biotin-labeled proteins to the cell surface; recycling was arrested by cooling the cells back down to 4°C. The recycled biotinylated proteins were stripped of biotin by using the MESNA buffer, and the cells were lysed to determine the biotinylated AC1 remaining in the cytosol. Biotinylated AC was quantitated using western blotting.

## Quantitative real-time PCR

Total RNA was extracted from the heart of wild-type and TRIP-Br1$^{-/-}$ mice by using an RNA extraction kit (Takara Minibest universal RNA extraction kit), and cDNA was synthesized using Superscript II reverse transcriptase from Invitrogen. Quantitative PCR (qPCR) amplifications of various genes were performed using SYBR Green (Takara) and an ABI 7500 fast real-time PCR machine (Applied Biosystems); GAPDH was used as a normalization control. After initial denaturation (30 s, 95℃), amplification was performed using 40 cycles of 5 s at 95℃ and 34 s at 60℃. All qPCR data represent means ± SEM. The sequences of the primers used for qPCR are listed below:

AC1: 5′-CCTCGCACTTACTGGTCACA-3′ and 5′-AACCCACGATGTCTGCAAAC-3′;
GAPDH: 5′-CTTGTCATCAACGGGAAGCC-3′ and 5′-CATGAGCCCTTCCACAATGC-3′.

## Forskolin and isoproterenol treatment of mouse macrophages

Peritoneal macrohphages induced by Brewer thioglycollate medium were harvested from pathogen-free adult mice in cold PBS, as previously described (*Misra and Pizzo, 2005*). After washed once with PBS and suspended in Roswell Park Memorial Institute (RPMI) medium with 10% FBS, macrophages harvested from one mouse were placed in a 6-well plate and incubated in a humidifiled CO$_2$ (5%) incubator for 2 hr at 37℃. The macrophages adhered to the culture plate were washed once with PBS to remove nonadherent cells, followed by incubation in RPMI medium overnight at 37℃. Subsequently, the macrophages were washed one with PBS and incubated with DMSO (as vehicle), 10 µM forskolin (with 100 µM IBMX to block phosphodiesterases), or 10 µM isoproterenol in the absence or presence of the PKA inhibitor H89 (10 µM) in RPMI medium without serum for 6 hr. The microphages were then lysed with RIPA buffer with 1 x protease inhibitor cocktail and analyzed by western blotting.

## Immunostaining and microscopy analysis of HeLa cells

For confocal microscopy of Hela cells transfected with 3HA-AC1, 48 hr after transfection, cells were fixed with 4% paraformaldehyde (PFA) for 10 min and then washed once with PBS. The cells were then permeabilized and blocked with 3% bovine serum albumin (BSA) in a 'PBST' buffer (PBS plus 0.1% Triton X-100) for 30 min before incubated with rabbit anti-HA (1:100 dilution) and mouse anti-TRIP-Br1 (1:100 dilution) antibodies overnight at 4℃. Next, the cells were washed thrice (5 min each time) with the PBST buffer, followed by incubation with Alexa Fluor 488 conjugated goat anti-rabbit IgG and Alexa Fluor 647 conjugated goat anti-mouse IgG secondary antibodies (both 1:500 dilution) for 1 hr. Finally, the cells were washed thrice (5 min each time) with the PBST and subject to confocal microscopy.

For confocal microscopy of untransfected HeLa cells, cells seeded on coverslips were washed with PBS, incubated with/without (for control) anti-AC1 antibodies (1:50 dilution) in Opti-MEM containing Lipofectamine2000 (1:10 dilution) for 2 hr at 37℃, and then recovered in DMEM with 10% (v/v) FBS for 3 hr at 37℃ (*Dalkara et al., 2004*; *Liang et al., 2015*). After fixation with 4% PFA in PBS, the cells were incubated at room temperature sequentially with biotin-conjugated anti-rabbit IgG secondary antibody (1:200 dilution) in 4% BSA in the PBST buffer for 1 hr, with NeutrAvidin (5 µg/mL, Pierce, cat#31000) for 1 hr, and then with Atto 488 Biotin (20 µg/mL, Sigma-Aldrich, cat#30574) for 1 hr; the cells were washed thrice (10 min each time) with PBS between each step. To detect TRIP-Br1, the cells were then incubated with anti-TRIP-Br1 antibody (1:200 dilution) in 4% BSA in the PBST for 1 hr, followed by incubation with Alexa Fluor 647 conjugated goat anti-mouse IgG antibody (1:200 dilution, Abcam, ab150115) and DAPI (1 µg/mL, Sigma-Aldrich) for 40 min. The sample was examined under a confocal microscope (Leica STED TCS SP5 II).

For stochastic optical reconstruction microscopy (STORM) analysis, HeLa cells seeded on No.1 18 mm round coverslips were washed twice with PBS, fixed with 4% PFA in PBS for 1 hr, permeabilized with the PBST for 10 min, and blocked with 4% BSA in PBS for 1 hr. The cells were then incubated with anti-AC1 and anti-TRIP-Br1 primary antibodies for 3 hr and with Alexa Fluor 647 conjugated goat anti-mouse IgG (1:200 dilution) and preadsorbed Alexa Fluor 750 conjugated goat anti-rabbit IgG (1:200 dilution, Abcam, ab175733) antibodies for 1.5 hr. The sample was kept in PBS until imaging. The sample was examined using STORM in an imaging buffer containing: 200 mM Tris-Cl (pH 9.0),10% (w/v) glucose, 5 U/mL glucose oxidase type VII (Sigma-Aldrich, cat# G2133), 57 µg/mL catalase (Sigma-Aldrich, cat#C3515), 2 mM cyclooctatetraene (Sigma-Aldrich, cat#138924), 25 mM TCEP

(Sigma-Aldrich, cat#64654–7), 1 mM ascorbic acid (Sigma-Aldrich, cat#A92902) and 1 mM methyl viologen (Sigma-Aldrich, cat#856177). Images were analyzed using the image analysis software Fiji.

## Statistics

Protein bands in western blots were quantified using ImageJ software. All data are expressed as means ± SEM. n denotes the number of independent biological replicates. Unless indicated otherwise, Student's two-tailed $t$ test was used for statistical analysis, and $p < 0.05$ was considered statistically significant.

## Acknowledgements

We thank Drs. R Madhavan and H B Peng for helpful discussion and suggestions, Dr. M Serrano for providing TRIP-Br1 knockout mice, Drs. D Storm, R Reed, Z Vogel, K Chung, and P Jost for kindly providing plasmids, Ka-Lun So for technical support, and Nicole Hsu for graph preparation. This work was supported by the Hong Kong Research Grants Council grant GRF660913 (to PH).

## Additional information

### Funding

| Funder | Grant reference number | Author |
| --- | --- | --- |
| The Hong Kong Grants Council | GRF660913 | Pingbo Huang |

The funders had no role in study design, data collection and interpretation, or the decision to submit the work for publication.

### Author contributions

WH, XYu, KL, Conceptualization, Formal analysis, Investigation, Writing—original draft; ZL, XYa, Formal analysis, Investigation; YS, XL, WKL, YD, XCa, Investigation; XCh, Investigation, Writing—original draft; HS, Resources, Writing—review and editing; PH, Conceptualization, Resources, Formal analysis, Supervision, Funding acquisition, Validation, Investigation, Writing—original draft, Project administration, Writing—review and editing

### Author ORCIDs

Pingbo Huang, http://orcid.org/0000-0002-4560-8760

### Ethics

Animal experimentation: All animal procedures were approved by the University Committee on Research Practices at the Hong Kong University of Science and Technology (the ethics protocol number 2014028).

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
