## [Decision Letter]

Thank you for submitting your article "The complex of TRIP-Br1 and *XIAP* ubiquitinates and degrades multiple adenylyl cyclase isoforms" for consideration by *eLife*. Your article has been reviewed by three peer reviewers, one of whom, Volker Dötsch (Reviewer #1), is a member of our Board of Reviewing Editors and the evaluation has been overseen by Jonathan Cooper as the Senior Editor.

The reviewers have discussed the reviews with one another and the Reviewing Editor has drafted this decision to help you prepare a revised submission.

Summary:

In this manuscript, the authors describe extensive biochemical studies revealing a *XIAP*/TRIP-Br1-dependent mechanism for regulating cellular protein levels of adenylyl cyclases. The authors show several lines of evidence supporting that TRIP-br1 functions as an adaptor of *XIAP* E3 ligase for AC ubiquitination and degradation. The authors propose that *XIAP*/TRIP-Br1-mediated degradation of ACs can serve as a negative-feedback mechanism for controlling homeostasis of cAMP signaling. Most experiments were done with AC1. Interaction of TRIP-br1 with ACs and *XIAP*/TRIP-Br1-dependent AC ubiquitination and AC protein levels were shown in several other AC isoforms. TRIP-Br1 knockout mice and *XIAP*-deltaRING mice exhibit increased AC protein levels, consistent with the authors' model. The mutant mice also show AC-related physiological and behavioral phenotypes.

Essential revisions:

1) The conclusion for Figure 2 and Figure 2 would be better supported by including quantification analysis.

2) The ubiquitination assay in Figure 4 seems to mainly consist of one major band instead of the usual smear as seen for example in B. Any explanation? In this figure the expression level of TRIP-Br1 does not seem to increase with the overexpression of the flag tagged protein. Should the effect of overexpression and the resulting reduction of AC1 levels not be larger?

3) K27 modifications are rather unusual. For degradation K48 would be the preferred one. The type of ubiquitination is linked to the E2. Is there any indication of which E2 is involved?

4) Connected to the point above: A complete model would have to consider the binding and interaction of the E2 enzyme. Where would it interact with in the model in Figure 9?

Mouse work:

5) In Figure 7—figure supplement 1, the authors show a battery of behavioral analyses and conclude that knocking out TRIP-Br1 causes anxiety-like behavior. However, the behavioral phenotypes could be more complex than simply being interpreted as anxiety-like disorder. For example, forced swimming test is conventionally used for depression or despair. Nesting test has not been generally adopted for anxiety disorder. The authors should provide more convincing justification in order to conclude an anxiety-like disorder or be more conservative in describing this apparent mood disorder phenotype.

---

## [Author Response]

*Essential revisions:*

*1) The conclusion for Figure 2 and Figure 2 would be better supported by including quantification analysis.*

As suggested, the quantification of Figure 2 is now included in the revised Figure 2.

*2) The ubiquitination assay in Figure 4 seems to mainly consist of one major band instead of the usual smear as seen for example in B. Any explanation? In this figure the expression level of TRIP-Br1 does not seem to increase with the overexpression of the flag tagged protein. Should the effect of overexpression and the resulting reduction of AC1 levels not be larger?*

Depending on the conditions of SDS-PAGE, such as gel concentrations and electrophoresis/separation time, the pattern of ubiquitinated AC1 may appear different. The experiment in original Figure 3 used 10% gel (vs. 8% gel in other experiments; see the molecular markers) and relatively short running time and therefore ubiquitinated AC1 appeared more condensed, but still smeared, especially in the last two lanes. Nevertheless, most of the ubiquitinated AC1 are between 170 and 130 kD in both 8% and 10% gel.

It has been reported by various labs that FLAG tag masks the antigen of protein specific antibodies in western blotting for unclear reasons. In our case, FLAG-tagged TRIP-Br1 was barely detected by our anti-TRIP-Br1 antibody, unless the protein sample was boiled and added with 200 mM DTT (Figure 10). However, boiling would make AC1 aggregate. Therefore, the protein sample in Figure 4 was not boiled, and FLAG-tagged TRIP-Br1 was hardly detected by anti-TRIP-Br1 antibody. The original purpose to display the signals detected by anti-TRIP-Br1 antibody in Figure 4 was mainly to show that endogenous TRIP-Br1 was not affected by overexpression of *XIAP*. This panel was removed in the revised figure to avoid confusion.

Overexpression of transient FLAG-tagged TRIP-Br1 seemed to result in average 50% reduction in AC1 (see the large collective sample size in Figure 5, including wild-type and mutant AC1). 30 to 70% reduction (Figure 4) could be fluctuations around 50% because of the relative sample sizes.

In addition, we added a new panel (Figure 4) to further support the data in Figure 4 and other panels in Figure 4. Figure 4 shows that deleting the RING domain of *XIAP* abolished the effect of TRIP-Br1 knockout in the mouse brain, further fortifying the notion that *XIAP* and TRIP-Br1 act in concert.

Author response image 1.TRIP-Br1-FLAG was detected by TRIP-Br1 antibody.HeLa cells were seeded in a 60-mm dish the day before the transfection. 5 ug TRIP-Br1-FLAG or control vector were transfected into the HeLa cells. After 48 h culture, cells were lysed by RIPA buffer and the cell lysate was boiled at 95°C for 10 min and added with 200 mM DTT before subject to western blotting by anti-FLAG and anti-TRIP-Br1 antibodies separately. 12% gel was used for better separation of endogenous (arrowhead) and FLAG-tagged TRIP-Br1.**DOI:**
http://dx.doi.org/10.7554/eLife.28021.021

*3) K27 modifications are rather unusual. For degradation K48 would be the preferred one. The type of ubiquitination is linked to the E2. Is there any indication of which E2 is involved?*

We thank the reviewers to bring up this point. Although K27 linkages are indeed less abundant than K48 linkages, recent studies seem to suggest that K27 linkages are much more abundant than previously expected (9% K27 vs. 29% K48 linkages in the yeast) [1]. Numerous studies indicate that K27 linkage is involved in proteolytic functions [1-4].

UbcH5A, B, and C, which share 92% identity, have been functionally demonstrated to serve as the E2s of XIAP [5-9]. In our in-vitro ubiquitination assays, we used UbcH5B as the E2 of *XIAP* (Figure 5). Importantly, UbcH5A or UbcH5B reportedly catalyses formation of K27 linages [2, 10]. Limited studies suggest that *XIAP* may physically interact with several other E2 enzymes [11, 12] but whether they serve as the E2s of *XIAP* is not known or rejected [9].

We added several sentences and relevant references in the revised paper to address this issue.

*4) Connected to the point above: A complete model would have to consider the binding and interaction of the E2 enzyme. Where would it interact with in the model in Figure 9?*

Previous studies show UbcH5B binds to the Ring domain of XIAP [7, 6]. We added UbcH5B in the model in Figure 9.

*Mouse work:*

*5) In Figure 7—figure supplement 1, the authors show a battery of behavioral analyses and conclude that knocking out TRIP-Br1 causes anxiety-like behavior. However, the behavioral phenotypes could be more complex than simply being interpreted as anxiety-like disorder. For example, forced swimming test is conventionally used for depression or despair. Nesting test has not been generally adopted for anxiety disorder. The authors should provide more convincing justification in order to conclude an anxiety-like disorder or be more conservative in describing this apparent mood disorder phenotype.*

As suggested, we used a more conservative description: “anxiety-like behaviour” is replaced by “mood disorder, especially despair-like behaviour”.

1. Xu, P. *et al.* Quantitative Proteomics Reveals the Function of Unconventional Ubiquitin Chains in Proteasomal Degradation. *Cell* 137, 133–145 (2009).

2. Xia, T. *et al.* The four and a half lim-only protein 2 (FHL2) activates transforming growth factor β (TGF-β) Signaling by regulating ubiquitination of the E3 ligase arkadia. *J. Biol. Chem.* 288, 1785–1794 (2013).

3. Tamara A. Potapova, John R. Daum, Kendra S. Byrd, and G. J. G. Fine Tuning the Cell Cycle: Activation of the Cdk1 Inhibitory Phosphorylation Pathway during Mitotic Exit. *Mol. Biol. Cell* 82, 327–331 (2009).

4. Birsa, N. *et al.* Lysine 27 ubiquitination of the mitochondrial transport protein miro is dependent on serine 65 of the parkin ubiquitin ligase. *J. Biol. Chem.* 289, 14569–14582 (2014).

5. Damgaard, R. B. *et al.* Disease-causing mutations in the XIAP BIR2 domain impair NOD2-dependent immune signalling. *EMBO Mol. Med.* 5, 1278–1295 (2013).

6. Mace, P. D. *et al.* Structures of the cIAP2 RING domain reveal conformational changes associated with ubiquitin-conjugating enzyme (E2) recruitment. *J. Biol. Chem.* 283, 31633–31640 (2008).

7. Nakatani, Y. *et al.* Regulation of ubiquitin transfer by XIAP, a dimeric RING E3 ligase. *Biochem. J.* 450, 629–638 (2013).

8. Takeda, A.-N. *et al.* Ubiquitin-dependent regulation of MEKK2/3-MEK5-ERK5 signaling module by XIAP and cIAP1. *EMBO J.* 33, 1784–1801 (2014).

9. Suzuki, Y., Nakabayashi, Y. & Takahashi, R. Ubiquitin-protein ligase activity of X-linked inhibitor of apoptosis protein promotes proteasomal degradation of caspase-3 and enhances its anti-apoptotic effect in Fas-induced cell death. *Proc. Natl. Acad. Sci. U. S. A.* 98, 8662–7 (2001).

10. Nucifora, F. C. *et al.* Ubiqutination via K27 and K29 chains signals aggregation and neuronal protection of LRRK2 by WSB1. *Nat. Commun.* 7, 11792 (2016).

11. Markson, G. *et al.* Analysis of the human E2 ubiquitin conjugating enzyme protein interaction network Analysis of the human E2 ubiquitin conjugating enzyme protein interaction network. 1905–1911 (2009). doi:10.1101/gr.093963.109

12. Hu, S. & Yang, X. Cellular inhibitor of apoptosis 1 and 2 are ubiquitin ligases for the apoptosis inducer Smac/DIABLO. *J. Biol. Chem.* 278, 10055–10060 (2003).